# The copy number and mutational landscape of recurrent ovarian high-grade serous carcinoma

Philip Smith [1,19], Thomas Bradley [1,19], Lena Morrill Gavarró [1], Teodora Goranova[1], Darren P. Ennis[2], Hasan B. Mirza [2], Dilrini De Silva[1], Anna M. Piskorz[1], Carolin M. Sauer [1], Sarwah Al-Khalidi[1], Ionut-Gabriel Funingana [1,3], Marika A. V. Reinius [1,3], Gaia Giannone[2], Liz-Anne Lewsley[4], Jamie Stobo[4], John McQueen[4], Gareth Bryson[5], Matthew Eldridge [1], The BriTROC Investigators*, Geoff Macintyre [1,6], Florian Markowetz [1,20], James D. Brenton [1,3,20] ✉ & Iain A. McNeish [2,20] ✉

The drivers of recurrence and resistance in ovarian high grade serous carcinoma remain unclear. We investigate the acquisition of resistance by collecting tumour biopsies from a cohort of 276 women with relapsed ovarian high grade serous carcinoma in the BriTROC-1 study. Panel sequencing shows close concordance between diagnosis and relapse, with only four discordant cases. There is also very strong concordance in copy number between diagnosis and relapse, with no significant difference in purity, ploidy or focal somatic copy number alterations, even when stratified by platinum sensitivity or prior chemotherapy lines. Copy number signatures are strongly correlated with immune cell infiltration, whilst diagnosis samples from patients with primary platinum resistance have increased rates of *CCNE1* and *KRAS* amplification and copy number signature 1 exposure. Our data show that the ovarian high grade serous carcinoma genome is remarkably stable between diagnosis and relapse and acquired chemotherapy resistance does not select for common copy number drivers.

Ovarian high grade serous carcinoma (HGSC) is marked by ubiquitous *TP53* mutation[1], chromosomal instability (CIN), extensive copy number alterations[2–4] and marked inter- and intra-patient genomic heterogeneity. This complexity has prevented effective precision medicine strategies and the only molecular classification utilised in clinical practice is identification of tumours with defective homologous recombination (HR).

Clinically, response rates to first line platinum-taxane therapy are high (65% by imaging criteria, 85% by CA125 criteria[5]). However, nearly all patients subsequently relapse and the probability of response to further platinum-based chemotherapy decreases with each exposure, with the vast majority of patients eventually acquiring fatal chemotherapy resistance. Beyond rare revertant mutations in *BRCA1/2*[6], and loss of *BRCA1* promoter methylation[7], the drivers of recurrence

[1]CRUK Cambridge Institute, University of Cambridge, Cambridge, UK. [2]Ovarian Cancer Action Research Centre, Department of Surgery and Cancer, Imperial College London, London, UK. [3]Cambridge University Hospitals NHS Foundation Trust, Cambridge, UK. [4]CRUK Glasgow Clinical Trials Unit, Institute of Cancer Sciences, University of Glasgow, Glasgow, UK. [5]Department of Histopathology, Queen Elizabeth University Hospital, Glasgow, UK. [6]Centro Nacional de Investigaciones Oncológicas, Madrid, Spain. [19]These authors contributed equally: Philip Smith, Thomas Bradley. [20]These authors jointly supervised this work: Florian Markowetz, James D. Brenton, Iain A. McNeish. *A list of authors and their affiliations appears at the end of the paper.
✉e-mail: james.brenton@cruk.cam.ac.uk; i.mcneish@imperial.ac.uk

and resistance remain unclear. While the time interval since last platinum chemotherapy still remains the most useful predictor of subsequent treatment response, assessing the prevalence of temporal heterogeneity and evolution, and understanding how divergent evolution could underlie acquired resistance in HGSC is of great importance.

The recent development of copy number signatures provides a robust framework to quantify different types of CIN and to assign its extent and origins[8,9]. We previously described seven copy number signatures in HGSC, patterns of copy number change that were prognostic and statistically associated with specific mutational processes. For example, signature 1 was associated with mutations in RAS/MAPK pathway and poor overall survival, whilst signatures 3 and 7 were positively prognostic and significantly associated with defective homologous recombination[8].

We established BriTROC-1, a UK-based ovarian cancer translational study, to investigate the acquisition of resistance in women with HGSC by collecting tumour biopsies from women with relapsed HGSC. We previously demonstrated that obtaining tumour biopsies in relapsed HGSC is safe and feasible, and that these biopsies yield sufficient DNA for genomic analyses[10].

Here, we use this well-annotated cohort to show that the ovarian high grade serous carcinoma genome is remarkably stable between diagnosis and relapse and acquired chemotherapy resistance does not select for common copy number drivers. We also utilise BriTROC-1 samples to identify *CCNE1* amplification, *KRAS* amplification and CN signature 1 exposure as markers present at time of diagnosis that predict early, platinum-resistant relapse. Finally, we show that copy number signatures are strongly correlated with immune cell infiltration.

## Results

### Patients and samples

276 patients with relapsed ovarian high grade serous carcinoma (HGSC) were recruited between January 2013 and September 2017 from 14 UK centres. Clinical characteristics are summarised in Table 1 and Tables S1–4, and the study scheme in Fig. 1A. 209 patients were classified as platinum-sensitive and 67 as platinum-

resistant and the median time from diagnosis to enrolment was 31.5 months (range 10–284, IQR 21–56). The median number of lines of prior chemotherapy was 1 (IQR 1–2) for platinum-sensitive patients and 2 (IQR 1–2) for platinum-resistant patients. Germline *BRCA1/2* status was known for 98/276 patients at time of enrolment: 22 had known pathogenic *BRCA1* mutations and 14 pathogenic *BRCA2* mutations. All treatment before and following study enrolment was at the discretion of treating oncologists (Table S5 and supplementary data 1 and 2), and median overall survival following enrolment was 35.6 months for the platinum-sensitive patients and 11.5 months for the platinum-resistant cohort (Fig. 1). REMARK diagrams are presented in Fig. S1.

### Germline and somatic short variant analyses

Germline variants in key genes in the homologous recombination repair pathway were assessed in 228/276 patients (Fig. S2). Pathogenic mutations in *BRCA1* and *BRCA2* were identified in 25 (11%) and 14 patients (6%) respectively. Pathogenic mutations in non-*BRCA* homologous repair deficiency (HRD) genes were identified in four other patients, one each of *RAD51B*, *RAD51C*, *RAD51D*, *PALB2* and *BRIP1* (0.5% mutation rate for all). Somatic sequencing on samples from 264 patients identified a pathogenic *TP53* mutation in 252 (95%) (Fig. S3, supplementary data 3), whilst the somatic mutation rate for *BRCA1* and *BRCA2* was estimated as 1 and 5% respectively. Comparison of matched tumour sample pairs (*n* = 134 pairs) showed very close concordance between diagnosis and relapse, with new mutation events being rare (Fig. 2). No revertant mutations were identified in *BRCA1* or *BRCA2* in relapse samples.

### Copy number alterations between diagnosis and relapse cohorts

There was strong concordance between diagnosis and relapse copy number changes (Fig. S4). Subtracting the median copy number profile at diagnosis from that at relapse for all pairs generated a flat profile with only 1.8% of bins (1531/83,607) showing significant alteration (Fig. 3A). Moreover, the median gain was only 0.09 copies across these regions (IQR 0.05–0.10). The significantly altered bins overlapped with 322 protein coding genes, of which 59 were identified as cancer-related[11]. Gene ontology enrichment[12] analysis identified 111 significantly overrepresented biological processes after false discovery rate correction, one of which was regulation of MAP kinase activity ($q = 0.036$). However, reactome[13] analysis did not identify an overrepresentation of any given pathway or biological process in genes showing altered copy number.

We did not identify a co-ordinated difference in purity, ploidy, or copy number segments between diagnosis and relapse, including when stratifying by platinum sensitivity (Figs. S5–S7). We identified nine patients who showed ploidy changes between diagnosis and relapse (Fig. S6; increased in 7, decreased in 2). However, these nine patients did not differ from the remainder of the cohort in age, platinum sensitivity or lines of prior chemotherapy ($p = 0.053$; Mann–Whitney U test, $p = 0.69$; Fisher's exact test, and $p = 0.86$; Mann–Whitney U test, respectively). The total number of copy number events and features[8] were also broadly consistent between diagnosis and relapse (Figs. S8–9). We also observed no significant differences in chromosomal arm or cytoband copy number events. A higher amplification rate at two cytobands was identified at diagnosis compared to relapse, but neither remained significant after multiple testing correction (Fig. S10).

We next assessed focal amplification and deletion of 18 genes that are frequently altered in HGSC[3,14] (Table S6) and found no significant changes between diagnosis and relapse. The rate of *KRAS* amplification increased from 3.3% (4/122) at diagnosis to 9.4% (12/127) at relapse, although this did not reach statistical significance ($p = 0.07$; Fisher's exact test) (Fig. 3B). When only paired samples were analysed, we also found no significant

**Table 1 | BriTROC-1 patient demographic and disease characteristics**

| Characteristic | Platinum-sensitive (N = 209) | Platinum-resistant (N = 67) | Total (N = 276) |
|---|---|---|---|
| Median age at study entry (range), years | 66 (31–85) | 63 (24–81) | 65 (24–85) |
| Median time since diagnosis (range), months | 32.5 (10.2–284.2) | 23.8 (5.1–184.2) | 31.5 (5.1–284.2) |
| Histology, N (%) | | | |
| High grade serous | 200 (95.7) | 66 (98.5) | 266 (96.4) |
| Grade 3 Endometrioid | 6 (2.9) | 0 | 6 (2.3) |
| Carcinosarcoma | 1 (0.6) | 0 | 1 (0.4) |
| Missing | 2 (1.0) | 1 (1.5) | 3 (1.1) |
| **Prior treatment** | | | |
| Median number of regimens (range) | 1 (1–5) | 2 (1–12) | 1 (1–12) |
| 1, N (%) | 148 (70.8) | 19 (28.4) | 167 (60.5) |
| 2, N (%) | 39 (18.7) | 29 (43.3) | 68 (24.6) |
| 3, N (%) | 12 (5.7) | 8 (11.9) | 20 (7.2) |
| 4, N (%) | 5 (2.4) | 3 (4.5) | 8 (2.9) |
| >4, N (%) | 3 (1.4) | 7 (10.4) | 10 (3.6) |
| Data missing | 2 (1.0) | 1 (1.5) | 3 (1.1) |

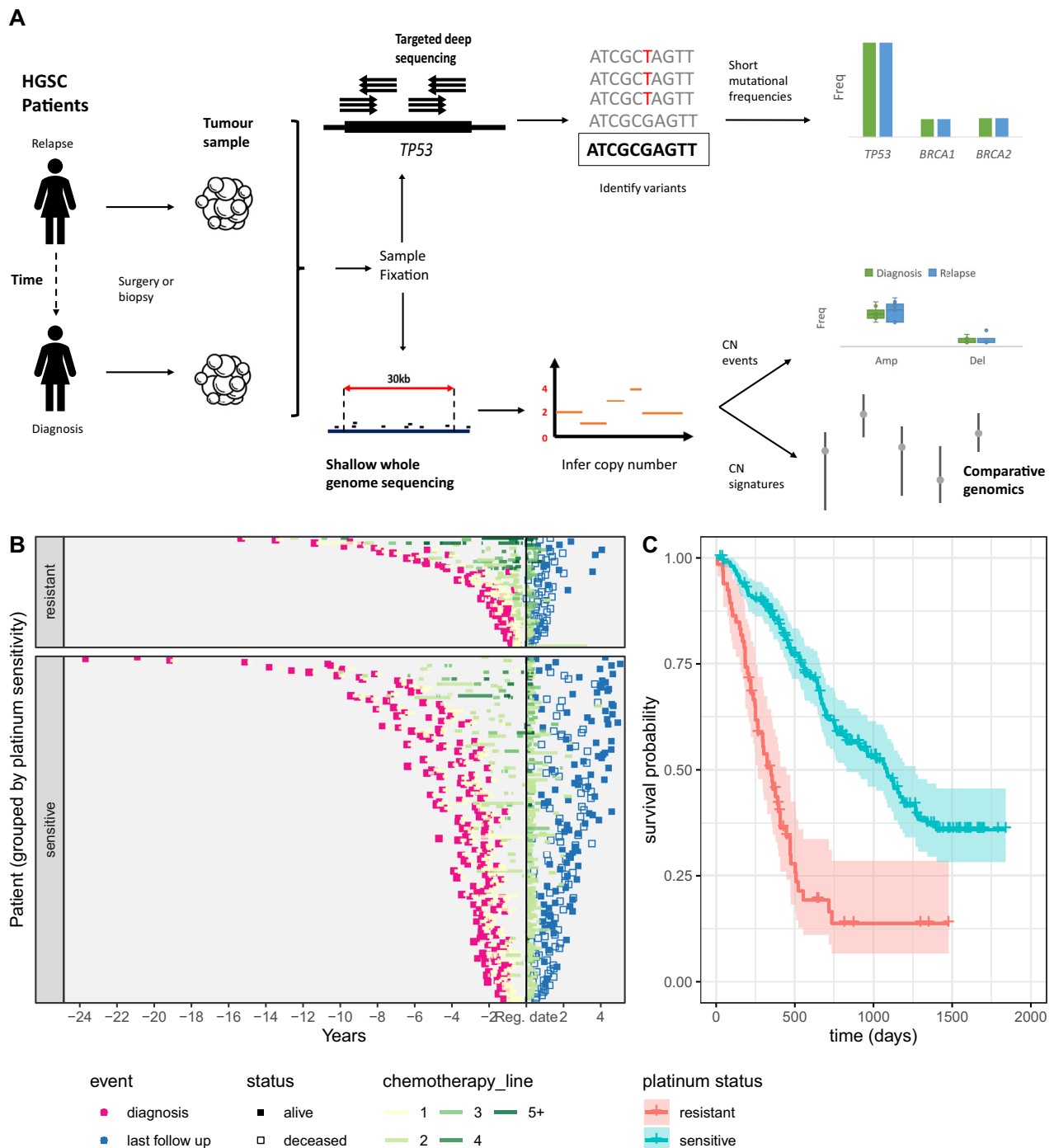

**Fig. 1 | BriTROC-1 Study scheme, recruitment timelines and overall survival.**
**A** Visual representation of study recruitment and data analysis pipeline. A workflow for the procedures, experiments and analyses conducted as part of this study. Patient samples were processed in parallel for short variant and copy number analysis. **B** Timeline representing the clinical course and outcomes of BriTROC-1 patients ($n = 269$). Each row represents a study participant; black vertical line represents the time of recruitment. Initial primary tumour diagnosis date is shown in pink and last follow up date at study end is shown in blue. Participants who died are shown as an unfilled point. Orange segments represent chemotherapy treatments and timescale of treatments, each subsequent treatment course is shown up to four. Treatment courses of 5 or more are aggregated. **C** Kaplan–Meier survival analysis for all patients enrolled into BriTROC-1 from the point of study entry stratified by platinum status. Crosses indicate right-censored data. The shaded areas indicate log-transformed upper and lower 95% confidence intervals computed from the standard error of the estimator of cumulative hazard for each condition.

difference in absolute copy number counts of these genes, including *CCNE1* (Fig. S11). Rates of amplification or deletion of these genes at relapse did not increase with increasing lines of prior therapy, in either paired or unpaired analyses (Fig. S12), nor when stratified by platinum status (sensitive vs resistant).

Absolute *CCNE1* gene copy number was marginally lower in relapse samples from platinum-resistant patients and *BRCA1* marginally higher in relapse sensitive patients ($p = 0.03$ and $p = 0.02$, Mann–Whitney U test) (Figs. S13–S14), though this was not significant after multiple testing corrections.

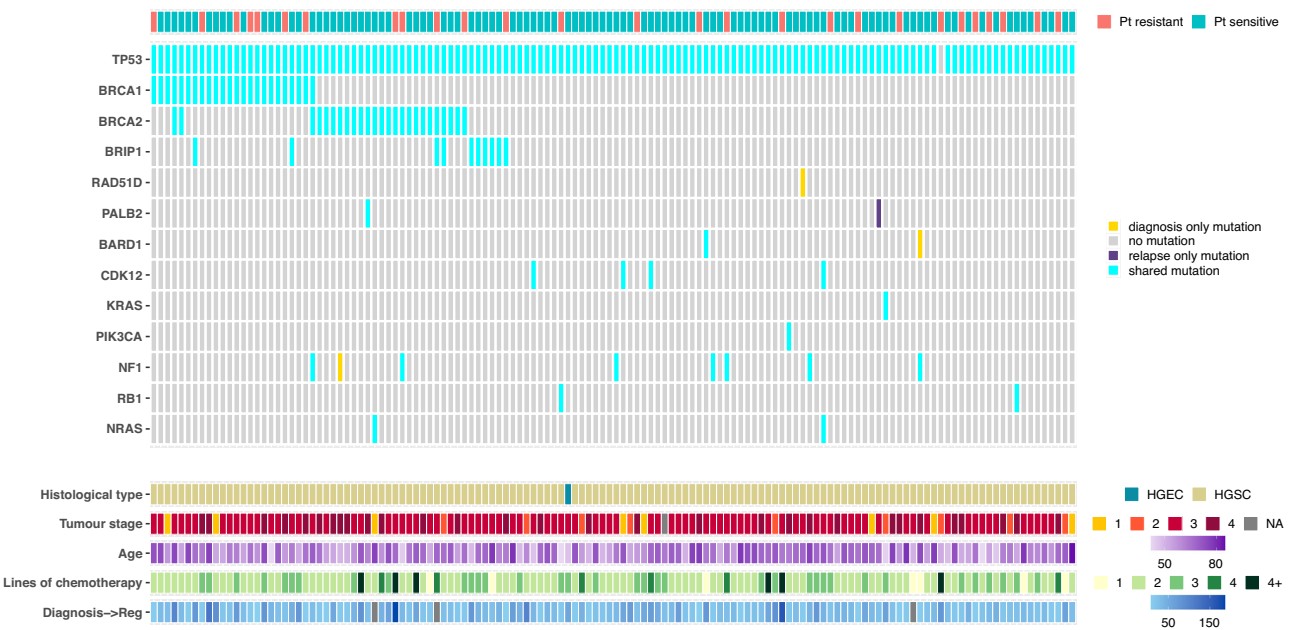

**Fig. 2 | Paired short-variant analysis from 134 diagnosis/relapse sample pairs.** Each column represents one patient and variants are not classified as either somatic or germline. Three cases had diagnosis-only mutations: patient 51 (*NF1* c.6643-3del), patient 101 (*RAD51D* p.Gln175Ter) and patient 163 (*BARD1* p.Val767Ala). Patient 139 had a relapse-only mutation in *PALB2* (p.Lys1163Glu). Abbreviations:

Pt - Platinum. HGSC - High grade serous carcinoma, HGEC - High grade endometrioid carcinoma. Tumour stage refers to stage at the time of diagnosis; Age refers to age at study entry; Lines of chemotherapy refers to the number of lines of chemotherapy prior to enrolment into BriTROC-1. Diagnosis-Reg denotes the interval (in months) between diagnosis and registration into BriTROC-1.

## Tumour heterogeneity

Intra-tumour heterogeneity (ITH) was estimated from the copy number profiles using the average segment distance from integer state across a given genome (supplementary methods). There was no statistically significant difference in ITH between diagnosis and relapse in either paired ($p = 0.27$, two-sided Wilcoxon signed-rank) and unpaired ($p = 0.18$, Mann–Whitney U test) analyses (Fig. S15). From paired samples, we found no difference in ΔITH, the ITH change between diagnosis and relapse, by platinum sensitivity ($p = 0.11$, Mann–Whitney U test), patient age at diagnosis ($p = 0.21$, Kendall's rank test), time between diagnosis and study registration ($p = 0.79$, Kendall's rank test), or number of prior lines of chemotherapy ($p = 0.85$, Mann–Whitney U test) (Fig. S16).

## Copy number signatures

Copy number signatures for all diagnosis and relapse samples were highly consistent with the initial analysis of 118 BriTROC-1 samples[8] (Fig. S17). We observed a small increase in exposure to signature 3 (s3) between diagnosis and relapse across the whole cohort (Fig. 4), that was not present in matched sample pairs (Fig. 5A, B). Given the compositional nature of copy number signature data (i.e. they add to 1 in all samples) and the fact that each patient had multiple samples, we designed a model to test for global differential abundance of copy number signature exposures between matched diagnosis and relapse groups (supplementary methods). Using a partial isometric Log-ratio (ILR) transformation and a generalised linear model with random effects, we identified a coordinated shift in s5 exposure, indicating differential abundance between diagnosis and relapse ($p = 0.003$, Wald test) (Fig. 5C). However, s5 is associated with chromothriptic-like events and subclonal changes that may partially reflect different fixation methods[15]. Most diagnosis samples were formalin-fixed, whilst relapse/study-entry samples were processed in a methanol-based fixative. We hypothesised that s5 was capturing differential fixation artefact rather than a true difference in mutational processes. Repeating the analysis whilst excluding s5 indicated no significant

difference in signature abundances between diagnosis and relapse samples ($p = 0.052$, Wald test).

## Copy number changes in different tissue sites in HGSC

To address whether metastasis to specific anatomical locations was associated with discrete genomic alterations, we first assessed the copy number of the eighteen key genes in different tissue sites across all samples (both diagnosis and relapse). Absolute copy number counts of *AKT1* and *MECOM* were significantly different between tissues ($p = 0.01$ & $0.005$, one-way ANOVA). Post-hoc testing and multiple testing correction identified statistically significant differences in *AKT1* copies between pelvic deposits and tissues classified as 'other' and *MECOM* copies in pelvic tissues compared to lymph nodes ($p = 0.006$ & $p = 0.006$, respectively, Bonferroni-corrected Tukey's test) (Fig. S18A). Rates of amplification and deletion were consistent across tissues after pairwise comparisons between tissue groups (Fig. S18B). ITH (Fig. S18C) and copy number signatures were also broadly consistent and were not statistically significant across tissue sites, except for copy number signature 1, which was statistically different between pelvic and lymph node tissues ($p = 0.04$, respectively, Bonferroni-corrected Tukey's test) (Fig. S18D).

When comparing diagnosis and relapse, we were restricted to abdominal, lymph node, pelvic, and peritoneal deposits due to sample number limitations. Absolute copy number counts in the eighteen key genes were consistent between diagnosis and relapse across different tissue sites and no significant differences in copy number counts were found. Similarly, copy number signatures and ITH, when stratified by diagnosis and relapse, were not statistically different across differing tissues of origin after multiple testing correction (Fig. S19).

## Patient-specific alterations between diagnosis and relapse HGSC

We assessed patient-specific gene alterations between diagnosis and relapse and determined if patient subgroups existed within the larger cohort-level analysis. Hierarchical clustering of the top 20% gene loci with the greatest variance in copy number between diagnosis and

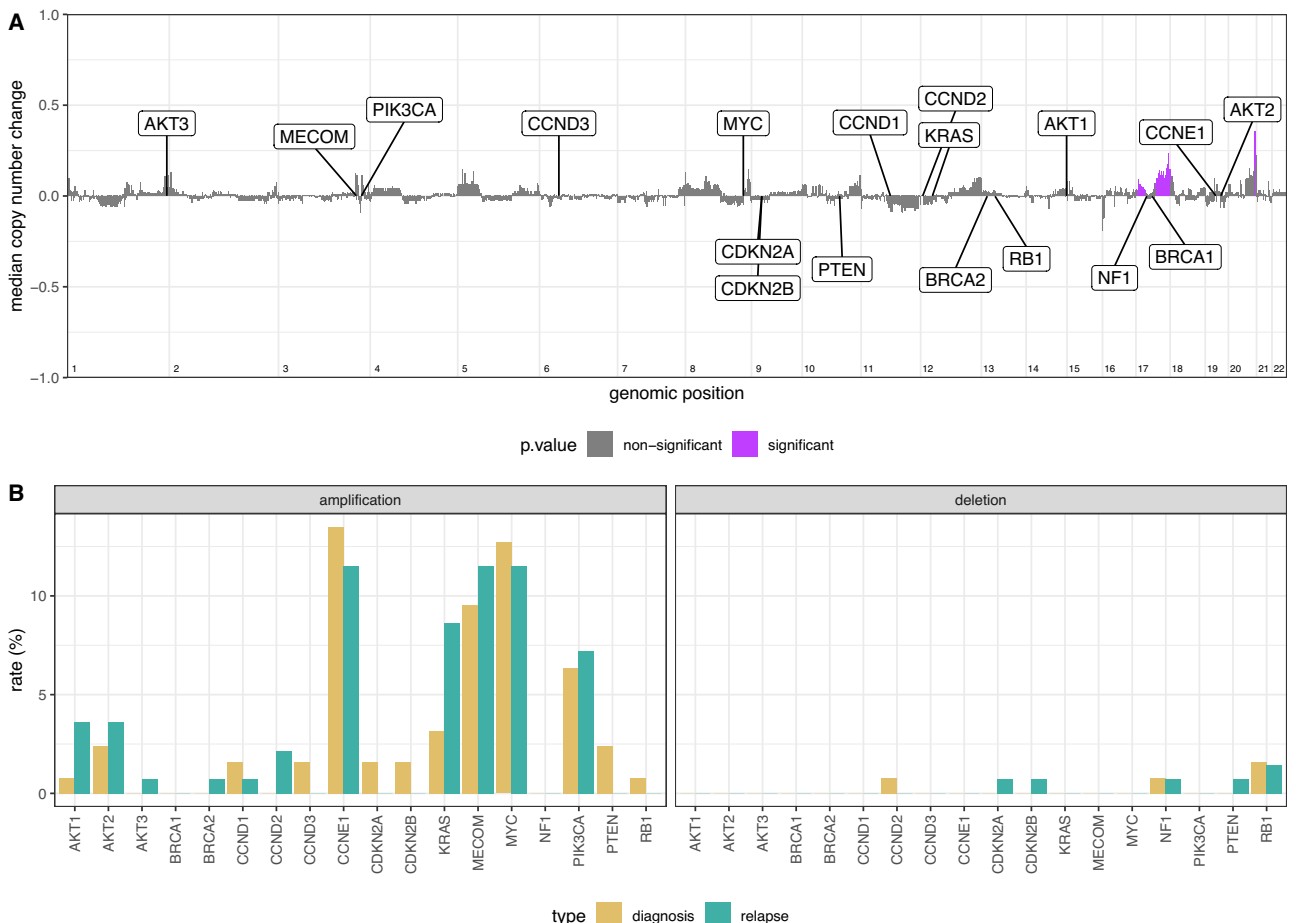

**Fig. 3 | Copy number alterations. A** Genome-wide subtraction plot to visualise copy number differences between diagnosis and relapse in 47 matched pairs. Positive values indicate an increase in copy number state and a negative value indicates a decrease in copy number state at relapse. Purple peaks represent genomic regions (30 kb bins) with significantly altered copy number between the diagnosis and relapse tumours (chromosome-specific FDR-corrected two-sided Mann–Whitney U test). Eighteen key and frequently altered genes in HGSC are highlighted. **B** Rates of focal amplification and deletion for the same eighteen genes ($n = 126$ and $n = 139$, diagnosis and relapse samples, respectively).

relapse identified a heterogeneous pattern of copy number alterations between patients. No obvious groupings were identified on the basis of specific genes or concordant changes with clinical features (Fig. S20). Focal genes that are frequently altered or of clinical relevance did not determine any clustering of patients. However, we did observe significantly correlated shifts in copy number changes between diagnosis and relapse in a subset of these genes, although the magnitude of copy number change was low (Fig. S21). Moreover, attempts to identify patient clusters using k-means clustering failed to identify any meaningful clusters (Fig. S22).

Looking specifically at the 18 frequently altered genes, we observed minimal differences between diagnosis and relapse across all paired samples when normalised for ploidy (Fig. S23). However, patient-specific analysis did demonstrate biologically interesting genomic alterations. For example, patients 65 and 37 demonstrated marked gains of *KRAS* at relapse; patient 36 had a 16 copy loss of *AKT2* and 6 copy loss of *NF1* at relapse, and patient 242 and 246 lost >10 copies of *CCNE1* between diagnosis and relapse (supplementary data 4 - patient vignettes). Cases with extreme copy gains and losses present across all gene loci were more frequent in platinum sensitive patients than resistant (Fig. S24).

**Primary platinum resistance**
Given the overall stability of genomic changes between diagnosis and relapse, we investigated genomes at diagnosis for features that

might be associated with poor outcome, specifically focussing on the 11 patients with primary platinum resistance (defined as relapse <6 months after completion of first-line treatment). We found higher rates of *CCNE1* and *KRAS* amplification at diagnosis in these patients compared to all other patients (58.3% vs 8.77%, $p = 0.002$; 25% vs 0.8%, $p = 0.05$, respectively, FDR-adjusted Fisher's exact) as well as higher absolute *CCNE1* gene copies (median 9.92 vs 2.73 copies, $p = 0.03$ FDR-adjusted Mann–Whitney U test) (Fig. 6A, Fig. S25). Intriguingly, diagnosis samples from primary platinum resistant patients also had higher absolute *BRCA2* copy number compared to all other diagnosis samples, although the magnitude of change was small (median 2.57 vs 1.98 copies, $p = 0.02$); Fig. S25). These differences remained evident at relapse (Fig. S25). The patients with primary platinum resistance also had significantly different CN signature exposures at diagnosis with significantly lower exposure to s3 and higher s1 exposure than samples from all other patients when comparing shifts in global signature abundance ($p = 0.003$, Wald test) (Fig. 6B, C).

**Immune correlations**
We investigated whether copy number alterations and chromosomal instability were associated with features of the tumour immune environment, using quantitative IHC for CD3 and CD8. There was statistically significant inter-marker correlation between CD3 and CD8 across tumour and stromal tissue independently ($\rho = 0.73, 0.73,$

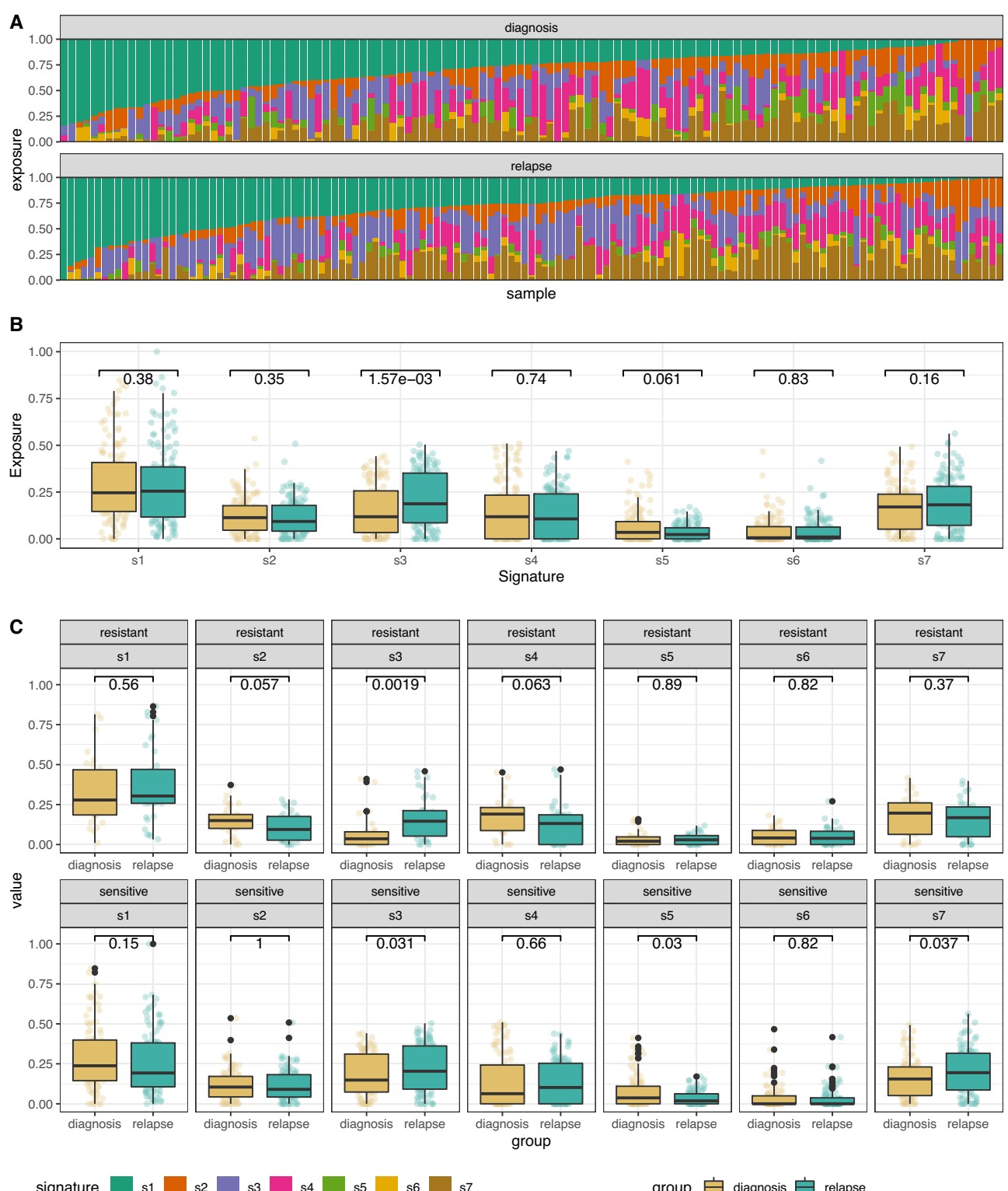

**Fig. 4 | Unpaired copy number signatures. A** Copy number signature spectrum across all samples. Stacked bar plots do not align between diagnosis and relapse groups due to different sample numbers ($n = 126$ and $n = 139$, diagnosis and relapse, respectively). **B** Boxplot demonstrating the unpaired copy number signature exposures between the diagnosis and relapse tumours ($n = 126$ and $n = 139$, diagnosis and relapse, respectively). Individual data points are overplotted. Statistics shown is a two-sided Mann–Whitney U test, without adjustments for multiple comparisons. **C** Boxplot demonstrating copy number signature distributions in unpaired copy number between diagnosis and relapse tumours, stratified by platinum status ($n = 30$, $n = 36$, $n = 96$, and $n = 103$, diagnosis resistant, relapse resistant, diagnosis sensitive, and relapse sensitive, respectively). Statistics shown is a two-sided Mann–Whitney U test, without adjustments for multiple comparisons. Boxplots show the lower and upper hinges corresponding to the first and third quartiles (the 25th and 75th percentiles). The whiskers extend from the hinge to the largest value no further than 1.5 × interquartile range from the hinge.

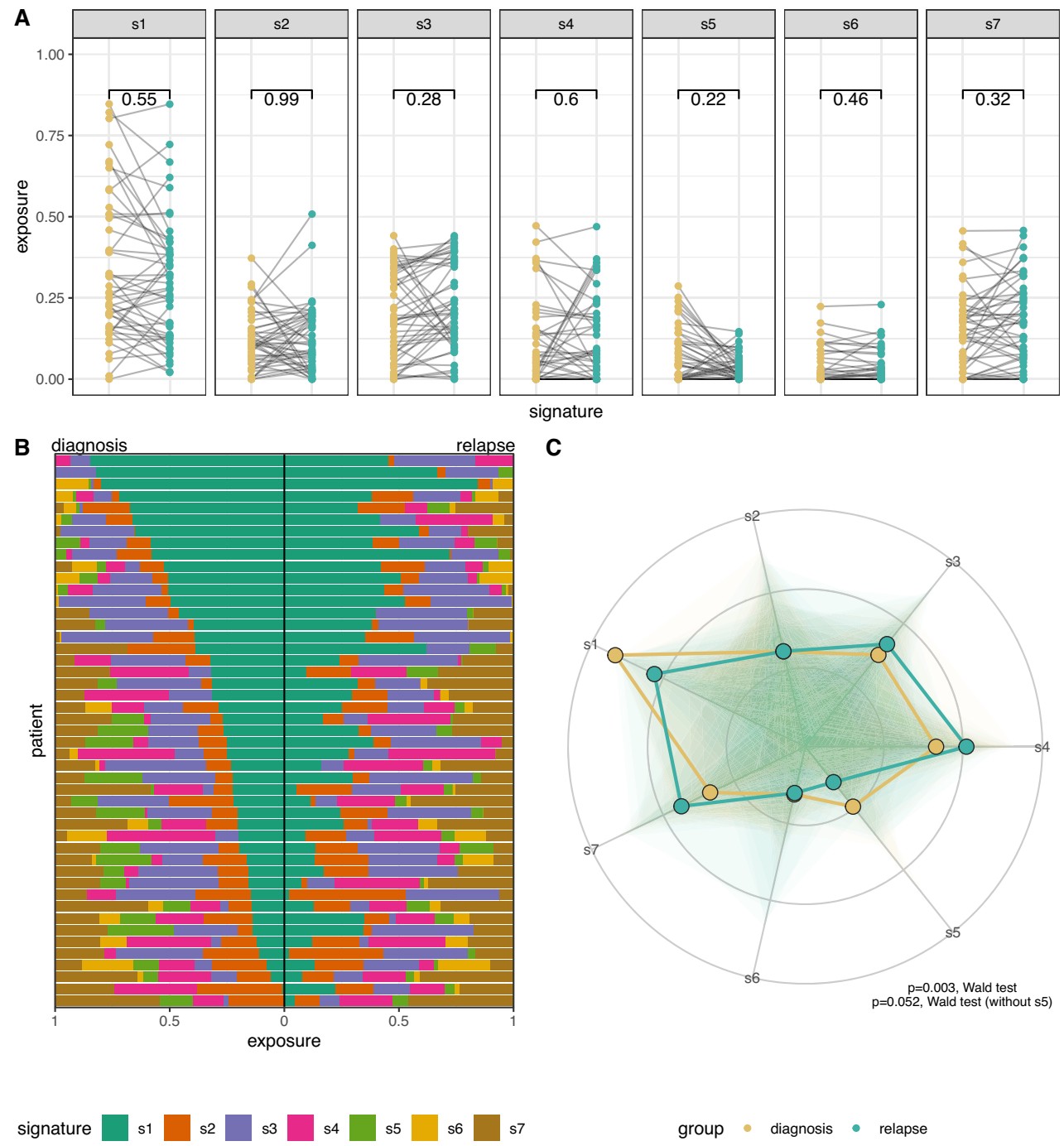

**Fig. 5 | Paired copy number signatures. A** Paired copy number signature exposures between the diagnosis and relapse tumours in 47 paired samples. **B** Copy number signature spectrum of paired copy samples. Each horizontal bar represents one patient, ranked by signature 1 exposure in the diagnosis sample (*n* = 47). Statistics shown is a two-sided Wilcoxon signed-rank test, without adjustments for multiple comparisons. **C** Radar plot of diagnosis and relapse copy number signature exposures. The distribution for all signature exposures for each comparison group is visualised using a shaded polygon. The radial points indicate the inverse ILR transformation of beta intercept and beta intercept + beta slope for each signature generated during signature modelling, for diagnosis and relapse groups, respectively. The difference in global signature abundance between diagnostic and relapse samples is significant when including signature 5 but non-significant without signature 5 (*p* = 0.003 & *p* = 0.052, respectively, two-sided Wald test).

and 0.72, Spearman's rank; all, stromal, and tumour tissue, respectively). CD3 and CD8 were shown to correlate strongly with copy number signatures. Specifically, s1 was negatively correlated with CD3 and CD8, s3 and s7 were both positively correlated with CD3 and CD8, and s6 was negatively correlated with CD3 (Fig. 7; Spearman's rank). These correlations were present in both stromal and tumour tissues, with a stronger signal identified in the stromal tissue.

## BRCA mutation status
Finally, to examine the effects of key mutational processes on the genome, we stratified patients into *BRCA*-mutant or -wild-type based on the presence or absence of a pathogenic germline or somatic variant in either *BRCA1* or *BRCA2*. Copy number alteration rates and gene numbers across 18 frequently altered genes were consistent between diagnosis and relapse samples when stratified by *BRCA* status. The rate of *MYC* amplification was greater in *BRCA*-mutant samples, although

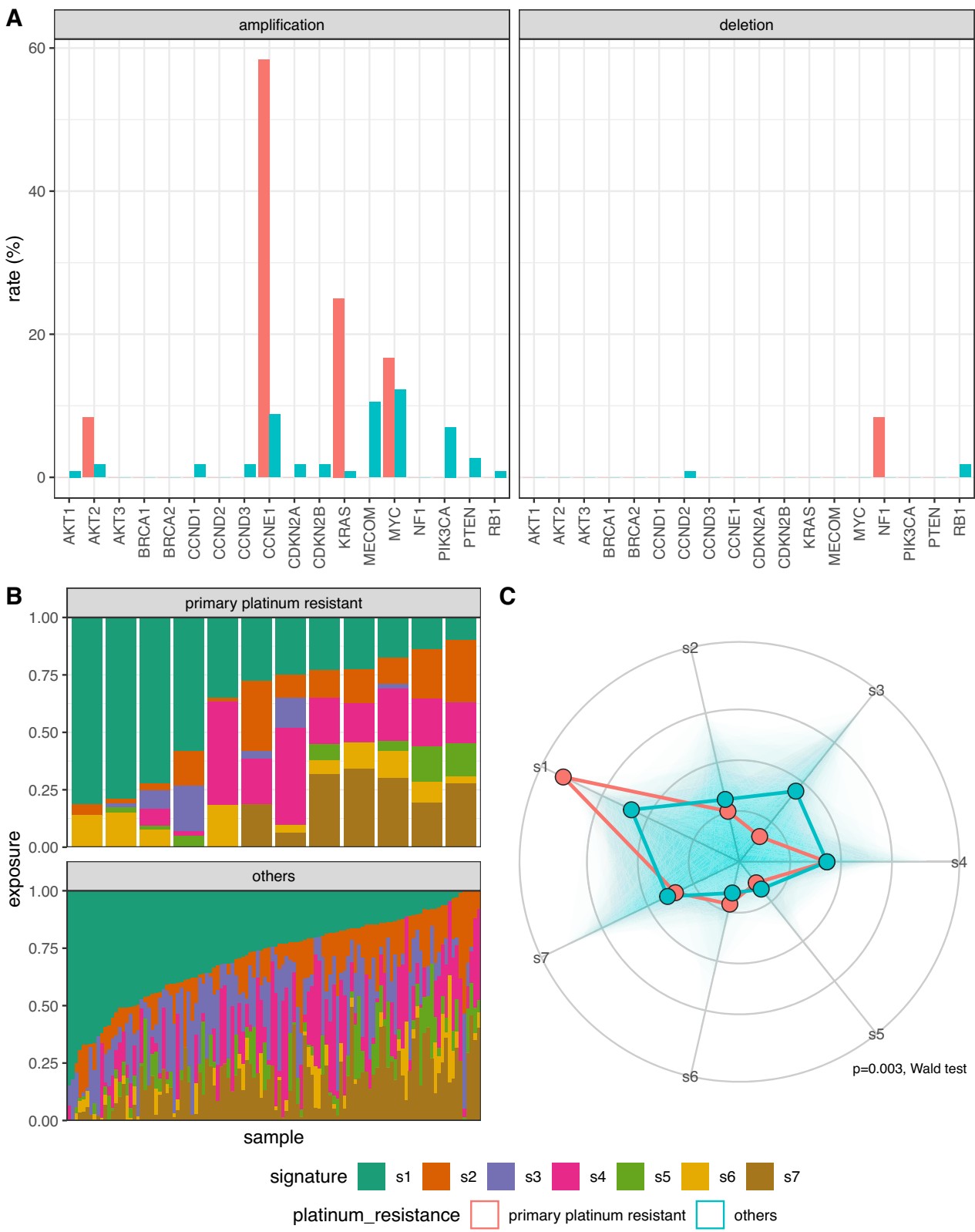

**Fig. 6 | Copy number signatures stratified by primary platinum-based treatment resistance. A** Copy number alteration rates for 18 frequently altered genes in diagnosis samples from BriTROC-1 patients relapsing within 6 months of completing first-line platinum chemotherapy ('primary platinum resistant') ($n = 12$) compared to all other patients ($n = 114$). **B** Copy number signature exposures and (**C**) Radar plot of primary platinum resistant cases and others copy number signature exposures. The distribution for all signature exposures for each comparison group is visualised using a shaded polygon. The radial points indicate the inverse ILR transformation of beta intercept and beta intercept + beta slope for each signature generated during signature modelling, for primary platinum resistant and all others, respectively. Differences in global abundance of copy number signatures between primary platinum resistant cases and all other samples was significantly different ($p = 0.003$, two-sided Wald test) driven by differences in s1 and s3.

**A** tumour

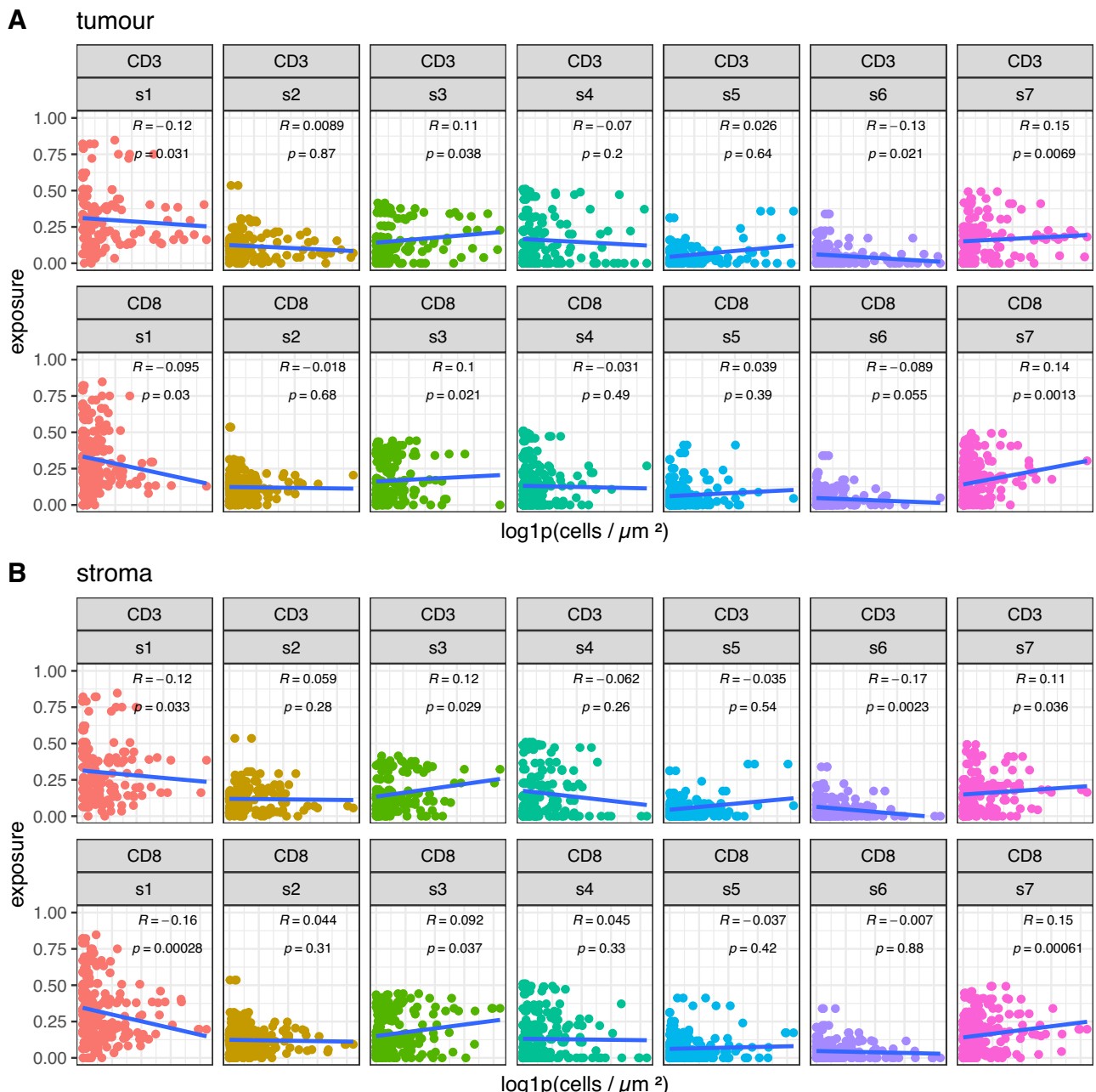

**B** stroma

**Fig. 7 | IHC-derived immune marker signature correlation plots.** Stratified correlations plot for IHC-derived CD3+ and CD8+ cell densities against copy number signatures. **A** Tumour and (**B**) Stroma. Solid blue lines are fitted linear regressions, *R* values are the Kendall tau correlation coefficient and associated *p* values for the significance of the given correlation. Calculated *p* values may not accurately estimate the strength of the correlation for each immune marker vs copy number signature due to the compositional nature of copy number signatures and intra-patient dependencies associated with the immune marker image data (CD3 = 158 & 58; CD8 = 239 & 90, IHC images and samples respectively). Statistics shown is a two-sided Kendall rank correlation coefficient test, without adjustments for multiple comparisons.

this did not reach statistical significance (Fig. 8A). *CCNE1* amplification was significantly more frequent in *BRCA*-wildtype, though not after multiple testing correction, ($p = 0.24$ FDR-adjusted Fisher's exact test; Fig. 8B), whilst *BRCA*-mutant samples had fewer absolute copies of *AKT3*, *CCND2*, and *CCNE1* ($p = 0.01$, $p = 0.01$, & $p = 0.013$, FDR-adjusted Mann–Whitney U test; Fig. 8C). When examining copy number signature differences by *BRCA* status, we were not able to identify a statistically significant shift in global abundance of copy number signatures between *BRCA*-mutant and wild-type ($p = 0.25$, Wald test). However, global shifts in signature abundance suggested an increase in s3 in *BRCA*-mutant cases as previously (Fig. 8D).

## Discussion

We used shallow whole genome sequencing and deep sequencing of a targeted gene panel to analyse samples from BriTROC-1, the largest prospective study yet of relapsed HGSC genomes. In a cohort of 276 cases, there are strikingly few recurrent changes between diagnosis and relapse. We identified only four cases with changes in single nucleotide variants/indels between diagnosis and study entry in a targeted panel of relevant HGSC genes, and no revertant mutations in *BRCA1* or *BRCA2* were detected in our population. Copy number profiles also showed minimal changes—we did observe selection for *KRAS* amplifications at relapse, although this did not reach statistical

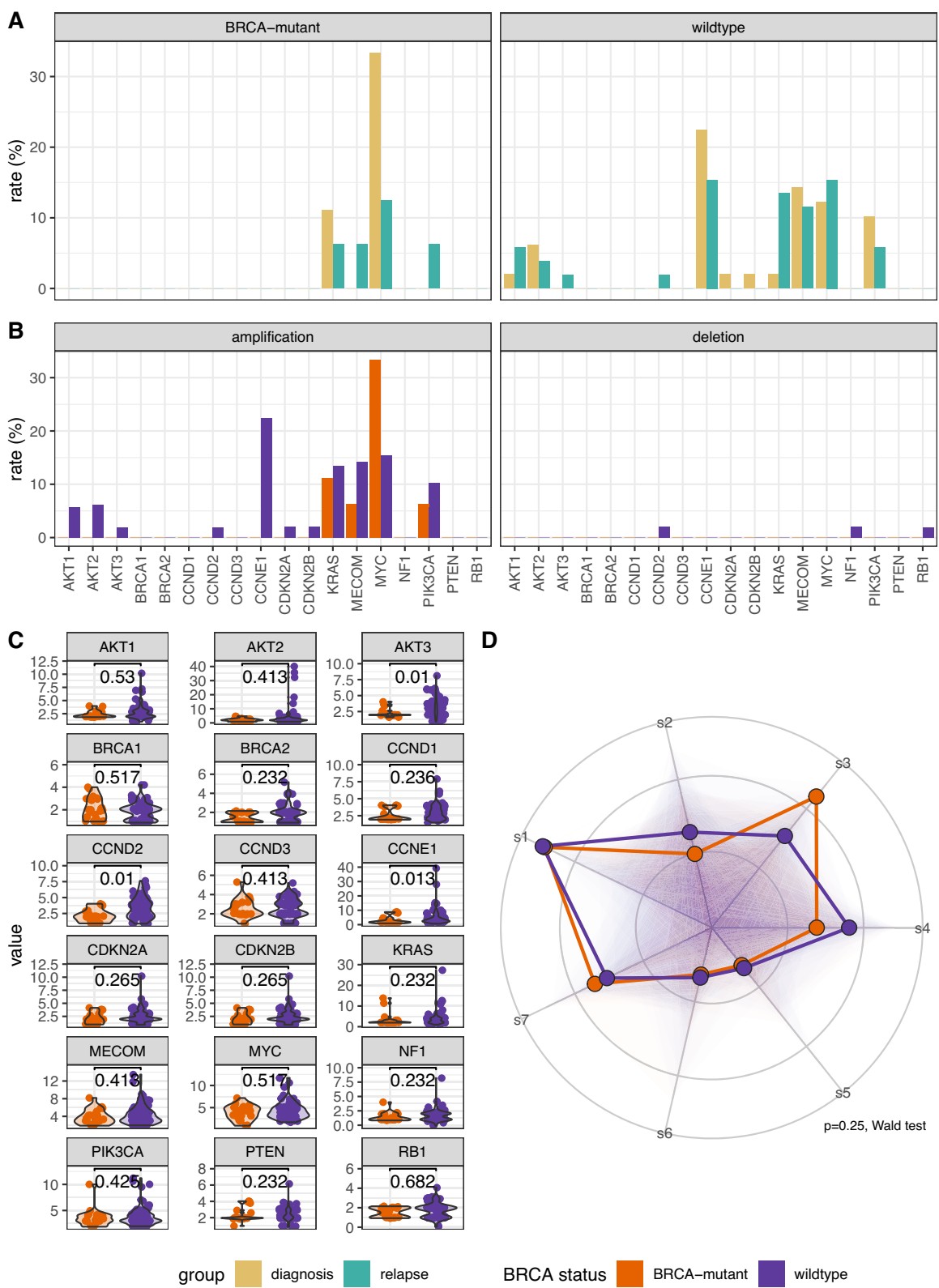

significance, suggesting that coordinated changes in driver CNA are infrequent. Copy number (CN) signatures did not show any statistically significant shift in exposures, suggesting that the mutational processes in HGSC either remain consistent or do not drive divergent patterns of CN alterations between diagnosis and relapse. We also did not identify recurrent changes in ploidy or intra-tumoural heterogeneity. These data strongly indicate that the major copy number features of HGSC,

as determined by our assays, are stable between diagnosis and relapse, and do not explain recurrence and acquired chemotherapy resistance.

The large size of the BriTROC-1 cohort and the stability of genomic changes allowed us to identify prognostic markers at diagnosis. In the patients who relapsed within 6 months of completing first-line chemotherapy, we found significantly higher rates of *CCNE1* and *KRAS* amplification at diagnosis than the remainder of the cohort. Although

**Fig. 8 | Copy number alterations and signatures stratified by BRCA status.**
**A** Copy number amplification rates for 18 frequently altered genes in paired samples with or without pathogenic alterations in *BRCA1/2*, stratified by diagnosis or relapse (*n* = 9, 16, 49, & *n* = 52, BRCA-mutant diagnosis and relapse, BRCA-wildtype diagnosis and relapse, respectively). **B** Copy number alteration rates for 18 frequently altered genes in paired samples alterations in BRCA or without (BRCA-mutant and wildtype, respectively), stratified by amplification or deletion event types (*n* = 25, & *n* = 101, BRCA-mutant and wildtype, respectively). **C** Absolute copy number state violin plots for the 18 frequently altered genes between paired BRCA and non-BRCA samples (*n* = 25, & *n* = 101, BRCA-mutant and wildtype, respectively).

Individual data points are overplotted. Statistics shown is a two-sided Mann–Whitney U test, without adjustments for multiple comparisons. **D** Radar plot of BRCA-mutant and wildtype copy number signature exposures. The distribution for all signature exposures for each comparison group is visualised using a shaded polygon. The radial points indicate the inverse ILR transformation of beta intercept and beta intercept + beta slope for each signature generated during signature modelling, for BRCA-mutant and wildtype, respectively. Differences in global abundance of copy number signatures between BRCA-mutant and wildtype samples was significantly different (*p* = 0.25, two-sided Wald test; *n* = 49 and *n* = 216, BRCA-mutant and wildtype, respectively).

*CCNE1* amplification has been identified previously as a poor prognostic feature in HGSC[16,17], we demonstrate here that amplification of non-mutated *KRAS* is also a marker of primary platinum resistance. In addition, CN signature 3 (s3), associated with defective HR[8], was significantly lower and s1 significantly higher in the primary resistant population at diagnosis. The absence of defective HR is associated with inferior outcome following platinum-based chemotherapy[18,19]. Exposure to s1, marked by low numbers of break points and larger segment sizes, is correlated with poorer survival, as expected of patients with primary platinum resistance, and s1 is anti-correlated with s3, a copy number signature associated with HRD.

*CCNE1* amplification rates were significantly lower in *BRCA1/2* mutated tumours compared to BRCA-wildtype, although the difference did not remain significant after multiple testing corrections. Although our previous analysis demonstrated an association of CN s3 and s7 with defective HR[8], here we found only a non-significant increase in s3 exposure between *BRCA*-mutant and -wildtype tumours, with no difference in global signature abundance. This may simply reflect low sample numbers (only 25 *BRCA*-mutated samples were of sufficient quality for signature analysis) and the model used, which evaluates global changes in signature composition rather than in individual signatures. In addition, the *BRCA*-wildtype cohort will include cases with other potential causes of HRD (e.g. *BRCA1* promoter methylation or mutations non-*BRCA* HR genes)[7], which may confound the comparisons.

We also show that CN signatures have strong associations with specific immune microenvironment at diagnosis. The presence of intra-tumoural T cells[20], in particular CD8+ cells[21], is strongly positively prognostic in HGSC. However, the tumour-autonomous drivers of immune cell infiltration remain elusive. Here, we found positive correlations between both CD3 and CD8 cell infiltration with increasing exposure to s3 and s7, and negative correlations with increasing s1. Previous studies have certainly identified an association between *BRCA1* loss and higher intra-epithelial CD8+ numbers[22,23], whilst tumours marked by fold-back inversions contain fewer CD8+ cells[24]. In primary platinum-resistant patients, we observed higher signature exposure to s1, which is strongly associated with breakage-fusion bridge mutational processes and overlaps with cases with fold-back inversions. Thus, our data extend known links between tumour genotype and immune phenotype and support the hypothesis that tumour-autonomous features strongly shape the immune microenvironment. However, more detailed analyses of the interplay between immune cell populations and tumour genotype in the BriTROC-1 cohort are ongoing.

Previous studies of HGSC evolution have examined multiple samples taken at primary surgery, revealing significant intra-patient heterogeneity with evidence of diverse metastatic processes and patterns of clonal expansion[25–27]. Analyses before and after neoadjuvant chemotherapy have also failed to observe recurrent chemotherapy-induced mutations, but did suggest copy number alterations, including *SIK2* amplification[28]. We did not identify any cases with *SIK2* amplification or any difference in *SIK2* absolute copy number (*SIK2* median copies; diagnosis = 2.04 vs relapse = 1.99, *p* = 0.41, Mann–Whitney U test). The OCTIPS consortium investigated 31

matched HGSC sample pairs (diagnosis and relapse, with 24/31 cases analysed at first relapse) with whole exome sequencing and SNP profiling[29]. Again, there were no consistent changes across pairs and the changes observed did not correlate with clinical characteristics. Similarly, there were no recurrent primary- or relapse-only unique somatic CNA in these pairs.

Broader studies examining the progression of tumours from diagnosis to relapse and/or metastasis have identified patterns of genomic alterations that are cancer-type dependent. Comprehensive analysis of genomic alterations in unpaired HGSC samples in the MSK-MET study also found no statistically different copy number alterations[30]. This is in stark contrast to other cancer types, including melanoma, which demonstrate distinct changes in SNV mutational signatures and large increases in copy number events, aneuploidy and whole genome duplications between diagnosis and end stage or metastatic disease[31,32]. Similarly, renal clear cell carcinoma demonstrates an evolutionary bottleneck followed by expansive increase in copy number alterations between primary and metastatic sites[33,34].

Our unpaired analyses suggested small but significant increases in s3 and s7 at relapse. The apparent increase of two signatures at relapse is consistent with our previous quantification of genome-wide LOH as a marker of defective HR[35]. In the ARIEL2 study, we observed a general increase in LOH between diagnosis and relapse that was sufficient to change classification from HR-proficient at diagnosis to HR-defective at relapse in ~15% cases[35]. Together, these data suggest an overall increase in CN damage as HGSC progresses, which may reflect both time-dependent and platinum-induced change.

There are several important caveats to our data. Firstly, BriTROC-1 enrolled women who had relapsed following prior therapy and who were well enough to undergo surgery or an image-guided biopsy, potentially biassing the study towards those with good prognosis. Nonetheless, overall survival following study enrolment was broadly consistent with previous clinical studies of chemotherapy in these populations[36–39], suggesting that our population responded as expected.

Secondly, we had to utilise routinely-collected formalin-fixed, paraffin-embedded diagnostic samples. This, combined with small biopsies at relapse, meant that the number of high quality matched sample pairs was limited, thereby reducing our statistical power and our ability to observe potentially important events. This limitation was most apparent in the analysis of CN signatures, where we observed a significant difference in global abundance of s5 between diagnosis and relapse: s5 is more prevalent in samples from FFPE sources[15] and we hypothesise that its differential abundance reflects a degree of fixation artefact.

Thirdly, we deliberately restricted our targeted sequencing panel given the low number of recurrent mutations seen in HGSC[3] and so were unable to identify mutations in other genes, quantify mutational burden or comment on mutational signatures. In addition, shallow WGS cannot assess genome-wide LOH and is subject to noise that is dependent on read depth and bin size[40]. It is thus possible we failed to observe recurrent rare events, such as patient-specific translocations in ABC transporter genes[14] or other structural variants[41] that can only be detected by deep WGS. Similarly, although we found no

coordinated copy number changes between diagnosis and relapse, some individual genomes showed marked changes. It is possible that these diverse patient-specific changes ultimately converge on a common phenotype of recurrence and resistance, but analysis of larger cohorts will be required to ascertain whether these changes are more frequent at relapse. Certainly, spatial transcriptomic analysis suggests the existence of discrete subclones with unique CN alterations within individual tumour sections that may be critical drivers of resistance[42].

Finally, a key potential driver of resistance is epigenetic change, which we did not examine here. Loss of *BRCA1* and *RAD51C* promoter methylation can drive platinum and PARP inhibitor resistance[7,43,44] whilst acquired methylation, in particular of *MLH1*, is seen in acquired platinum resistance models[45]. More broadly, RNAseq of matched pairs has previously identified changes in immune-related genes[46]. However, transcriptional subtypes of HGSC[47,48] derived from bulk analyses largely reflect immune cell composition and abundance of fibroblasts rather than intrinsic difference in malignant cells[49,50]. Overall, detailed single-cell analyses of matched pairs will be required to elucidate critical changes in copy number and gene transcription at relapse.

In summary, BriTROC-1 has allowed interrogation of genomes in relapsed HGSC, and revealed remarkable stability of copy number changes across time, despite the extreme complexity of the HGSC genome. These data suggest that common short variants and copy number alterations cannot explain the pattern of relapse and acquired resistance that is the major hallmark of this disease. Importantly, we identified new genomic events at diagnosis, including *KRAS* amplification and CN signature 1 exposure, that are associated with primary platinum resistance and may have predictive utility for patients receiving neoadjuvant chemotherapy.

## Methods

### Study conduct

Details of the BriTROC-1 study and the first 220 patients were reported previously[10]. Briefly, BriTROC-1 was funded by Ovarian Cancer Action (grant number OCA_006) and sponsored by NHS Greater Glasgow and Clyde. Ethics/IRB approval was given by Cambridge Central Research Ethics Committee (Reference 12/EE/0349) and the trial registration number is ISRCTN09180474. All patients provided written informed consent—this included specific consent to biopsy, access to archival material, use of biopsy and archival material (and ascites if present) for genomic studies, testing of germline DNA for *BRCA1/2* and other mutations and the use of clinical data for the research purposes. In addition, patients could optionally consent to a second biopsy upon disease progression and to be informed of germline *BRCA1/2* analysis results.

### Statistics & reproducibility

BriTROC-1 was originally powered to identify differences in the rates of defective homologous recombination in patients with platinum sensitive relapse, with a sample size 300. A total sample size of 300 (100 with platinum-resistant disease, 200 with platinum sensitive disease) provided power (>90%) to detect a 15% increase in the HRD status (based on the biopsies) in the sensitive group compared to the resistant group (assuming a 10% HRD rate in the resistant group) at the 5% two-sided level of statistical significance. The study recruited 276 patients, which reduced the power to 80%. No data were excluded from the analyses and there was no randomisation of participants—allocation to platinum-sensitive or platinum-resistant groups was based on clinical parameters and was designated by the treating oncologists. Experiments were not randomised. No blinding was performed. However, during primary analyses (germline and somatic variant calling; absolute copy number and copy number signature determination), researchers had no access to clinical information. Final analyses were performed with reference to clinical parameters.

### Patients

Between 16/JAN/2013 and 05/SEP/2017, the study enrolled patients with recurrent ovarian high grade serous or grade 3 endometrioid carcinoma who had relapsed following at least one line of platinum-based chemotherapy. Other histological subtypes were only allowed in patients with known deleterious germline *BRCA1* or *BRCA2* mutations. Patients were classified as platinum sensitive (relapse ≥6 months since last platinum-based chemotherapy) or platinum resistant (relapse <6 months since last platinum-based chemotherapy) by recruiting sites at the time of study registration. All patients had to have disease amenable either to image-guided or other interventional (e.g. endoscopy, bronchoscopy, or secondary debulking surgery) biopsy. Access to archival diagnostic formalin-fixed tissue samples, or snap frozen tumour material if available, was also required for patient registration. Overall survival was calculated from the date of enrolment to the date of death or the last clinical assessment, with data cut-off at 01/APR/2018. Full inclusion and exclusion criteria are listed in supplementary methods.

Patients underwent biopsy (at least two cores, 14–16 G biopsy needle) or secondary debulking surgery, with tumour samples fixed in methanol (UMFIX, TissueTek Xpress, Sakura)[15]. For patients undergoing secondary debulking or other interventional biopsies, 14–16 G cores or a 1 cm³ piece of macroscopically identified tumour tissue were taken. All samples were shipped within 24 h at ambient temperature to the University of Glasgow. There were no study-mandated therapies and all treatment after study entry was at the discretion of the treating oncologist.

### Tagged-amplicon sequencing

Normal and tumour DNA samples were assessed for single nucleotide variants and short indels using tagged-amplicon sequencing[51]. Paired-end sequencing was performed on either MiSeq and HiSeq 4000 Illumina platforms at 125 and 150 nt respectively.

A combination of different amplicon panels was used to assess mutational state. These amplicon panels contained the following genes: Panel 1 (*PIK3CA, EGFR, BRAF, PTEN, KRAS and TP53*), panel 6 (*APLF, BARD1, APTX, BRCA2, PARP2, FANCM, RAD51B, PALB2, RAD51D, BRCA1, RAD51C, PPM1D, BRIP1*) panel 10 (*TP53*), and panel 28 (*BRCA1, BRCA2, RAD51C, RAD51D, RAD51B, BRIP1, FANCM, PALB2, BARD1, CDK12, EGFR, PTEN, TP53, KRAS, BRAF, PIK3CA, CTNNB1, NF1, RB1, NRAS*). The amplicon panels and loci are provided in supplementary data 5.

### Amplicon read alignment

Alignment and post-alignment processing methods for sequenced amplicon reads are described in the supplementary methods.

### Germline variant calling

Germline short variants were called using CRUK-CI's ampliconseq pipeline (https://github.com/crukci-bioinformatics/ampliconseq; v0.7.2)[8] using GATK's HaplotypeCaller (GATK version 3.8-0-ge9d806836[52]) as the core variant calling algorithm and also using the octopus method (v0.7.2)[53]. Further details are in supplementary methods.

### Tumour sample variant calling

Variant calling on tumour samples was performed using the *cancer* calling mode of Octopus (v0.7.2)[53] with the exception of *TP53* variants, which were as reported previously[8,10,15]. Further details are in supplementary methods.

### Short variant functional annotation

All non-*TP53* variants were functionally annotated using Ensembl's variant effect prediction (VEP) pipeline[54] (v102.0). Variants were further refined using the molecular tumour board portal (MTBP)[55]. Variants labelled as 'benign' or 'likely benign' by MBTP were discarded. Further details are in supplementary methods.

## Shallow whole genome sequencing and alignment

Single-end 50 bp read length sequencing was performed with 0.1× coverage target. Fastq sequencing reads were aligned to GRCh37 (hs37d5) using bwa samse (version 0.7.17-r1188) and duplicates were marked using picard MarkDuplicates (version 2.9.5).

## Absolute copy number profile fitting

Relative copy number profiles for each sample were fitted using a bespoke absolute copy number pipeline (see code availability) from single-end shallow whole genome sequencing (~0.1× coverage). Initial relative copy number profiles were generated using a modified version of QDNAseq (supplementary methods)[40]. Relative copy number profiles were then fitted to an optimal ploidy and purity combination after GC and mappability correction, including a quantitative and qualitative quality control, to generate absolute copy number profiles using 30 kilobase bins (supplementary methods). A REMARK diagram is provided in Fig. S1. Copy number event calling thresholds and quality metric assignment of ploidy change patients are detailed in the supplementary methods.

## Intra-tumour heterogeneity

Intra-tumour heterogeneity was estimated for absolute copy number profiles using a methodology analogous to that published previously[56] with alterations described (supplementary methods).

## Copy number signature analysis

Copy number signatures were derived using the methodology described previously[8], utilising the pre-computed signature definitions to generate the signature exposures for all samples with available absolute copy number profiles. Copy number signatures were compared between groups using well established methodologies, as well as a statistical framework to determine global shifts in signature abundance between groupings described herein (supplementary methods).

## Immunohistochemistry

Quantitative immunohistochemistry (IHC) data were available for a subset of diagnosis samples. Methodology for the generation and normalisation of IHC data is described in detail in the supplementary methods.

## Reporting summary

Further information on research design is available in the Nature Portfolio Reporting Summary linked to this article.

## Data availability

All genomic data relating to the BriTROC-1 study are available via EGA under accession code EGAC00001000388. This also includes data from previous BriTROC-1 publications[8,10,15]. The genomic sequencing data are available under restricted access due to patient confidentiality. Access can be obtained by applying to the data access committee via EGA. It is expected that data will be available within 3 months of publication and there are no restrictions on the duration of access. The pre-processed single nucleotide variant and copy number data are available through a Zenodo data repository and can be downloaded here. The source data for the figures generated in this study are provided in the Source Data file provided with this paper. Source data are provided with this paper.

## Code availability

All code required to reproduce the analysis in this paper is freely available and linked at https://github.com/BRITROC/britroc-1-HGSOC-landscape which details the utilised analysis pipelines, copy number fitting pipelines, and data access links. This repository is citable using the https://doi.org/10.5281/zenodo.7942206 via Zenodo release tracking.

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

## Acknowledgements

This work was supported by Ovarian Cancer Action (grant number OCA_006 to IAMcN); the National Institute for Healthcare Research (NIHR) Imperial Biomedical Research Centre (grant number P77646 to IAMcN); the Wellcome Trust PhD programme in Mathematical Genomics and Medicine (grant number RG92770 to LMG); Cancer Research UK (grant numbers DRCQQR-Jun22\100005, A22905, A15601 and A26204 to JDB, FM and IAMcN); a European Society of Medical Oncology (ESMO) Translational Fellowship (to G.G.); the Beatson Cancer Charity and Hutchison Whampoa Limited. Infrastructure support was provided by CRUK/NIHR Experimental Cancer Medicine Centres at Imperial, Cambridge, Glasgow and other participating sites. We also thank the Biorepository, Bioinformatics, Histopathology, IT & Scientific Computing and Genomics Core Facilities of the Cancer Research UK Cambridge Institute and the Pathology Core at the Cancer Research UK Beatson Institute for technical support. The funders had no role in study design, data collection and analysis, decision to publish or preparation of the paper.

## Author contributions

Conception or design of the work; I.A.M., J.D.B., F.M., G.M. Patient recruitment; R.M.G., C.G., R.K., G.H., R.E., A.C., S.S., A.W., M.L., M.H., H.G., C.F., E.B., A.M. Data acquisition; P.S., T.B., T.G., D.P.E., G.B., H.M., A.M.P., S.A.K., C.S., I.F., M.A.V.R., G.G. Clinical study co-ordination; L.A.L., J.S., J.M. Data analysis and interpretation, P.S., T.B., L.M.G., T.G., H.M., M.E., D.D.S., S.A.K., C.S., I.F., M.A.V.R. Paper writing: P.S., T.B., J.D.B., I.A.M. Paper revision; P.S., T.B., I.A.M. All authors reviewed and approved the paper.

## Competing interests

G.M., F.M., A.M.P. and J.D.B. are founders and shareholders of Tailor Bio Ltd. The remaining authors declare no competing interests.

## Additional information

## The BriTROC Investigators

R.M. Glasspool[7], C. Gourley[8], R. Kennedy[9], G. Hall[10], R. Edmondson[11], A. Clamp[12], S. Sundar[13], A. Walter[14], M. Hall[15], H. Gabra[2], C. Fotopoulou[2], E. Brockbank[16], A. Montes[17] & M. Lockley [18]

[7]Beatson West of Scotland Cancer Centre, Glasgow, UK. [8]University of Edinburgh, Edinburgh, UK. [9]Queen's University, Belfast, UK. [10]University of Leeds, Leeds, UK. [11]University of Manchester, Manchester, UK. [12]Christie Hospital Foundation NHS Trust, Manchester, UK. [13]University of Birmingham, Birmingham, UK. [14]Bristol Oncology Centre, Bristol, UK. [15]East and North Hertfordshire NHS Trust, Stevenage, UK. [16]Barts and the London NHS Trust, London, UK. [17]Guy's and St Thomas' NHS Trust, London, UK. [18]Barts Cancer Institute, Queen Mary University of London, London, UK.

