## [Peer Review File · Nature Communications]

REVIEWER COMMENTS

Reviewer #1 (Remarks to the Author): expertise in ovarian cancer genomics

The authors present an interesting study of copy number alterations in a large cohort of primary and post-treatment recurrences of high grade serious ovarian carcinoma, searching for genomic correlates of primary and acquired resistance to platinum-based therapy. While the patient cohort is remarkable and poised to reveal important associations, the authors observed a high degree of stability in copy number profiles across the longitudinal timepoints. They fairly comment about the limitations of the study, that would have greatly benefited from a matching transcriptional dataset. Nonetheless, the study confirms previously described associations between CCNE1 amplification and HRD status with primary resistance and sensitivity to platinum therapy. It also reveals a few potentially interesting novel genomic associations, such as KRAS amplification and primary resistance, while it does not provide any major insights into the possible causes of tumor relapse.

Overall, I do not believe that this study improves our understanding of the biology/genomics of ovarian cancer recurrence (as the title seems to anticipate). The focus on CN signatures, while probably dictated by the technical constraints linked to the available specimens, has clearly limited the investigation of the full spectrum of genomic changes that contribute to ovarian carcinoma origins and evolution. I do believe though that the patient cohort is quite remarkable and deserves to be described and that with some more focused analyses, particularly by taking into account the different tumor drivers at play (CCNE1 vs BRCA1/2, more detailed comment below), it could provide some interesting novel insights.

Major points.

- Copy number alterations between diagnosis and relapse cohorts. While the authors' analysis shows no/limited significant differences in copy number between the diagnosis and the relapse datasets, this analysis is designed to reveal recurrent events across the patient population. It would be interesting to see what the overall trend for copy number changes is for individual tumor pairs from the same patient donor. Some of this is addressed in subsequent figures, but a critical point is to analyze samples with respect to different genetic backgrounds. For example, the expectation is that the HRD-deficient tumors would continue to acquire large deletions and, if BRCA1 mutant, tandem duplications (although, based on the authors definition of copy number alterations, even if detected, tandem duplication would not be considered alterations in their analysis). These alterations would not be consistent across patients when considering specific genes/genomic regions, but the overall trends (i.e., size and gain vs loss) could reveal a consistent pattern.
- In fact, given that some of the only significant differences observed in the study revolve around CCNE1 amplification, KRAS amplification and s3 CN signature (i.e. HRD deficiency), patient/tumor stratification based on CCNE1, KRAs and BRCA1/2 status may reveal interesting patterns. At a minimum, it would be helpful to see increased s3 in BRCA1/2 mutant tumors as a validation of the CN signature concept.

- The authors have previously published papers defining and describing the concept of the copy number signatures and how they may be relevant to the biology of a cancer. Here they simply refer to those publications. However, the concept is still fairly novel, and a brief introduction to what these signatures represent would help understanding the results that are discussed. A brief mention of the previously reported associations between a CN signature and genetic features/mechanisms when the signature is found to be associated with a specific tumor group (i.e., s3 = HRD, s1 = BFB) would be very helpful (rather than having to wait for clarification in the discussion).

- In general, the study would benefit from a more streamlined data presentation: there's a large number of supplementary figures/panels, many of which are redundant and occasionally confusing (some examples are reported in the comments below, Fig. 7 is a prominent case).

Minor remarks.

- Please specify in the method section what the original data source for the Absolute copy number profile fitting protocol is (i.e., I imagine it's the shallow whole genome sequencing, but it is not mentioned in the text nor in the Supp methods).

- Method section. Please, present a more detailed description of the Tagged-amplicon sequencing protocol: Table S1 reports the amplicons, but it would be helpful to report in the method section the total number of genes that are assessed and, since the list is not long, a list of all of the Gene Symbols.

- Figure S15. Please add a legend with the color-code. It would also be helpful to have a number of samples for each graph (i.e., how many diagnosis and how many relapse samples are analyzed in the graph relative to 4 prior lines of chemotherapy?) Some of the differences may seem remarkable without this context, for example, it appears that ~50% of relapse samples in this category have amplification of the AKT1 gene vs. 0% of the diagnosis samples. Is this just a fluke due to low numbers?

- Figure 4B. A p.value should be added to the boxplots.

- Figure S23. This figure is complex and without a statistical analysis highlighting the results worth noticing, it is very hard to interpret.

- Figure 5A. Please add 'NS' to indicate that the comparisons are not significant, if appropriate.

- Figure 5C. Without a p.value label, it is hard to recognize the shift in s5 as the significant feature in the figure (i.e., the shift in s1 seems just as pronounced). Also, since the authors eventually refute the biological nature of this shift, Figure 5C should be relegated to the supp data.
- Figure S24. Please add the significant p.values to the corresponding graphs.
- Figure 7. The term 'non-primary resistant' in the legend is confusing (same as in Fig. S31). Are these primary (i.e., at diagnosis) tumors that are not resistant, resistant tumors that are not primary, or any tumors that do not show primary resistance? I believe the first definition is correct, but it warrants clarification. Maybe better terms would be 'tumor with/without primary resistance'. Also, since primary resistance is defined as relapse <6 months after completion of first-line treatment, is it appropriate to classify relapse tumors as primary resistant, or should the distinction be limited to diagnosis tumors?
- Figure 7C, D and E. These graphs need some more detailed explanations. The text simply reports that 'patients with primary platinum resistance also had significantly different CN signature exposures at diagnosis with significantly lower exposure to s3 and higher s6 exposure than samples from all other patients'. While Fig. 7C does show these significant differences, they are not obvious in Fig. 7E (where s1 seems to show a larger delta) and it is unclear what Fig. 7D represents. Also increased s1 in patients with primary resistance is reported in the discussion but not in this corresponding result section, which adds to the confusion.
- Figure S31B. Please double check that the violin plots match the number of samples reported in the legend.

Reviewer #2 (Remarks to the Author): clinical expertise in ovarian cancer and treatment response

This report presents an extremely valuable real-world dataset of genomic analyses of initial tumor samples and biopsies at relapse from women with ovarian cancer. Surprisingly, the investigators identify limited differences in genomic parameters at the time of relapse. Overall this work is hypothesis generating, and it builds on prior analyses of ovarian cancer cohorts as described in the discussion. In particular, the identification of markers associated with primary platinum resistance present an opportunity to stratify patients for frontline treatment if validated clinically. Additional strengths of the study include the relatively tight timeframe during which patients were recruited and secondary samples obtained, the heterogeneity of treatment regimens that reflect standard clinical practice, and the paired analyses enabling an assessment of correlations between immune parameters and genomic alterations as well as differences in tumor profiles for individual patients. The inclusion of substantial

supplementary data - in particular individual patient vignettes - creates a resource that will continue to support research efforts in the field.

Additional information that would be helpful to guide interpretation of the data includes:

- Clarifying the specific make-up of subgroups analyzed for different parameters. Although 276 patients were enrolled, the analyses included much smaller cohorts ranging from 21 to 134 subjects. A table is provided in the supplemental data that tracks the number of patients included in each analysis but it's not clear why samples from only 47 patients were available for paired CN analysis or whether these subgroups differed significantly from the entire study cohort with respect to clinical characteristics.

- It would be helpful to have a separate statistical methods section included.

- Survival curves are included but this is not an interventional trial. With substantial differences (>10 yrs) in the time since diagnosis as illustrated in Fig 1B, a comparison of outcomes is difficult to interpret.

- The authors acknowledge the potential for bias in selecting patients healthy enough for biopsy/secondary debulking. It would be helpful to additionally note whether subjects were already scheduled for biopsy or secondary debulking by their treating oncologist, or whether this was done specifically for the purposes of this study.

- While differences in fixation methods are discussed as potential confounding factors for analyses, the age of the samples is not considered. Were older samples more likely to fail QC? Did this skew the cohorts analyzed?

- The title of the paper specifies HG serous cancers, but pts with non-serous cancers were also enrolled. Fig S1 suggests these cases were specifically excluded from some analyses. A justification for these decisions would be helpful to include in the methods.

- The individual vignettes of patients is a particularly rich source of information but a legend would be helpful - to explain the dashed red lines on the figures, to specify which biopsy was used for this study, etc. In addition, some cases appear to be missing data. For example pt 147 is classified as platinum resistant but she was treated with platinum-based chemotherapy after biopsy. Patient 132 had two debulking surgeries and a biopsy but it's not clear which sample(s) were included as the relapse sample.

Finally, several patients (157, 232, 187) had recurrence diagnosed but no treatment started for several months. This is surprising and may indicate there is missing treatment information.

Minor:

- Fig 1A has an arrow directed from the relapse figure to the figure at diagnosis? Fig 1B legend says orange blocks indicate chemo - these are green in the version provided.

Reviewer #3 (Remarks to the Author): expertise in statistical analysis of clinical trial data

“The genomic landscape of recurrent ovarian high grade serous carcinoma: the BriTROC-1 3 study”

The manuscript reported the results from the BriTROC-1 study which aims to investigate genes that are associated with recurrence and resistance in ovarian high grade serous carcinoma. The study is well designed, with its sequencing data gone through appropriate bioinformatic pipelines and statistical analysis. To seek genomic signal associated with innate and acquired resistance, the study investigated genome-wide mutation, copy number, selected genes and gene signatures on population level as well as on individual patient level (paired samples). A throughout set of analysis have been carried out, including comparisons guided by various prognostic clinical features, pathway-level enrichment analysis, and clustering to explore hidden patient sub-cohorts. I am convinced by the conclusion that HGSC has a stable genome, and it is unlikely the cause of acquired platinum resistance.

Comments:

1. The abstract needs more key numbers. I understand that the study did not find much genomic changes between diagnosis and relapse samples but the phrase such as “very strong concordance” is too vague for the readers to capture the key messages of the study.
2. The study included various data types and analysis, each comprised different numbers of patient samples. While the authors mentioned that the limited sample sizes are weak points of some analysis, I found it hard to keep track on sample sizes during reading. Please make sure that sample size information is properly included for each analysis in the main text, as well as corresponding figures, supplementary figures, and tables. One example would be on line 324, where the $p=0.02$ was achieved on a comparison between 7 patients and 19. Unfortunately these small sample sizes were not mentioned in the main text.
3. In line 207, I can not align the mutation events reported in Fig 2 with those in Fig S4, S5. Please double check.

4. Line 327, please describe clearly which site has significantly higher S1, as it is not obvious from visual in the Figure S24D.

5. Line 375, the association between CCNE1/KRAS amplification and the binary status of platinum sensitivity was assessed. Can you also show the association between the genes and time to relapse to enable better understand of the relationship between the genes and resistance?

6. Line 378, please double check the colour code of Figure S31-B, as the results reported in the main text can not be aligned with the figure.

RESPONSE TO REVIEWERS' COMMENTS

Reviewer #1: expertise in ovarian cancer genomics

The authors present an interesting study of copy number alterations in a large cohort of primary and post-treatment recurrences of high grade serous ovarian carcinoma, searching for genomic correlates of primary and acquired resistance to platinum-based therapy. While the patient cohort is remarkable and poised to reveal important associations, the authors observed a high degree of stability in copy number profiles across the longitudinal timepoints. They fairly comment about the limitations of the study, that would have greatly benefited from a matching transcriptional dataset. Nonetheless, the study confirms previously described associations between *CCNE1* amplification and HRD status with primary resistance and sensitivity to platinum therapy. It also reveals a few potentially interesting novel genomic associations, such as *KRAS* amplification and primary resistance,

Response: We thank the reviewer for these comments. We believe that this is the largest series of matched diagnosis/relapse samples in ovarian cancer, but we acknowledge in the manuscript the limitations of these samples.

We agree that transcriptional data would be hugely interesting but that lies beyond the scope of what we could achieve with small image-guided biopsies. Further work characterising the tumour immune microenvironment using imaging mass cytometry and gene expression analysis of archival samples from BriTROC-1 will be published as a separate study. However, it is also important to note that the most validated signature for gene expression in HGSC, PROTYPE (Talhouk et al Clinical Cancer Research 2020), does not encompass mutational processes and remains a weak classifier for outcome.

Overall, I do not believe that this study improves our understanding of the biology/genomics of ovarian cancer recurrence (as the title seems to anticipate). The focus on CN signatures, while probably dictated by the technical constraints linked to the available specimens, has clearly limited the investigation of the full spectrum of genomic changes that contribute to ovarian carcinoma origins and evolution. I do believe though that the patient cohort is quite remarkable and deserves to be described and that with some more focused analyses, particularly by taking into account the different tumor drivers at play (*CCNE1* vs *BRCA1/2*, more detailed comment below), it could provide some interesting novel insights.

Response: We politely disagree with the reviewer that this study does not improve our understanding of the biology of ovarian cancer. As stated above, we believe that this is the largest series of matched diagnosis/relapse samples in ovarian carcinoma, and, critically, we have studied the molecular consequences of mutational processes across different patients. Although other series, including Patch et al (Nature 2015) and Burdett et al (Nature Genetics 2023), have analysed samples at greater depth, their cohorts were much smaller and more limited (e.g. Burdett et al only included patients with mutations in *BRCA1*, *BRCA2* and *BRIP1*). By contrast, our study is informative of unselected, real world recurrent populations in standard oncology practice. Our data are clinically important as they show that recurrent genomic changes for major drivers are rare in relapsed HGSC and that signatures of mutational processes remain relatively invariant. This has not been shown in previous studies.

We address the specific questions raised by the reviewer (relating to *BRCA1/2*) below.

Major points.

- Copy number alterations between diagnosis and relapse cohorts. While the authors' analysis shows no/limited significant differences in copy number between the diagnosis and the relapse datasets, this analysis is designed to reveal recurrent events across the patient population. It would be interesting to see what the overall trend for copy number changes is for individual tumor pairs from the same patient donor. Some of this is addressed in subsequent figures, but a critical point is to analyze samples with respect to different genetic backgrounds. For example, the expectation is that the HRD-deficient tumors would continue to acquire large deletions and, if BRCA1 mutant, tandem duplications (although, based on the authors definition of copy number alterations, even if detected, tandem duplication would not be considered alterations in their analysis). These alterations would not be consistent across patients when considering specific genes/genomic regions, but the overall trends (i.e., size and gain vs loss) could reveal a consistent pattern.

Response: Thank you for raising this important point and the reviewer is correct that we would not consider a tandem duplication as an independent CNA. We previously started to address this question in supplementary Figure S25 in the original manuscript and have now further examined whether the predominant mutational processes, such as HRD, continue to pattern the genome consistent over time. We have investigated the copy number changes associated with BRCA mutational status, and also investigated the changes between diagnosis and relapse samples when stratified by BRCA mutation status. We specifically addressed whether there was continued acquisition of large deletions in HRD tumours as suggested by the reviewer, as well as trends of overall gains and losses over time. However, we were unable to demonstrate these changes convincingly. The figure below plots the percentage of the genome at diagnosis and relapse showing gains and losses, segregated by size. For gain and loss of segments between diagnosis and relapse, we could not identify a clear acquisition of "large" deletions in BRCA-mutated tumours at relapse.

We also assessed the number of these copy number segments in a paired approach and found high amounts of heterogeneity in the gain and loss of segments of various sizes, without any obvious patterns of

recurrence. This matches what we have described in the manuscript where additional copy number events at relapse are patient-specific, including the size and distribution of copy number events. Furthermore, we did not perform joint segmentation to allow breakpoint flexibility, which complicates the assignment of unique events acquired between diagnosis and relapse, particularly where patients can have multiple diagnosis or relapse samples allowing for many-to-many or many-to-one comparisons. Our future studies focusing on scDNA methodologies that should be able to address these questions more effectively.

- In fact, given that some of the only significant differences observed in the study revolve around CCNE1 amplification, KRAS amplification and s3 CN signature (i.e. HRD deficiency), patient/tumor stratification based on CCNE1, KRAs and BRCA1/2 status may reveal interesting patterns. At a minimum, it would be helpful to see increased s3 in BRCA1/2 mutant tumors as a validation of the CN signature concept.

Response: We have already published very extensive validation of the signature associations (Macintyre et al Nat Genet 2018, Drews et al Nature 2022) and the work here is wholly based on robust signatures described in our Nature Genetics manuscript.

As stated in the previous response, we have performed an analysis stratifying our sample set by *BRCA* status, which included an implementation of our signature comparison model for global changes in signature abundance, which we present as new Figure 8. This suggested a shift towards an increase in s3 in *BRCA*-mutant cases as previously, but this did not reach statistical significance. However, our sample number here was low (only 25 *BRCA*1/2 mutated samples yielded sufficient DNA for signature analysis), and the sample-patient structure of this dataset required a statistical analysis model capable of appropriately handling intra-patient correlations. As such, the model tests global changes in signature composition, rather than testing each signature independently (which does demonstrate significance for s3 in both diagnosis and relapse samples). Finally, the *BRCA*-wildtype cohort will include cases with HRD that arises through alternative mechanisms, including *BRCA*1 promoter methylation and mutation in non-*BRCA* HR genes, which may confound the comparisons.

- The authors have previously published papers defining and describing the concept of the copy number signatures and how they may be relevant to the biology of a cancer. Here they simply refer to those publications. However, the concept is still fairly novel, and a brief introduction to what these signatures represent would help understanding the results that are discussed. A brief mention of the previously reported associations between a CN signature and genetic features/mechanisms when the signature is found to be associated with a specific tumor group (i.e., s3 = HRD, s1 = BFB) would be very helpful (rather than having to wait for clarification in the discussion).

Response: Thank you. We have now expanded the introduction so that mutational signatures are more accessible for the general reader.

- In general, the study would benefit from a more streamlined data presentation: there's a large number of supplementary figures/panels, many of which are redundant and occasionally confusing (some examples are reported in the comments below, Fig. 7 is a prominent case).

Response: We have improved the captions and layout of the main figures. However, the large number of supplementary figures reflects the size of our cohort and the important corroborating analyses despite this we have curated and reduced the number of listed supplementary figures. Nonetheless, we have reduced the overall number of supplementary figures.

Minor remarks.

- Please specify in the method section what the original data source for the Absolute copy number profile fitting protocol is (i.e., I imagine it's the shallow whole genome sequencing, but it is not mentioned in the text nor in the Supp methods).

Response: We have added clarification to the methods section

-
- Method section. Please, present a more detailed description of the Tagged-amplicon sequencing protocol: Table S1 reports the amplicons, but it would be helpful to report in the method section the total number of genes that are assessed and, since the list is not long, a list of all of the Gene Symbols.

Response: We have clarified the genes included in the tagged-amplicon sequencing panels in the Methods section with a full list of amplicons and loci in Table S1.

-
- Figure S15. Please add a legend with the color-code. It would also be helpful to have a number of samples for each graph (i.e., how many diagnosis and how many relapse samples are analyzed in the graph relative to 4 prior lines of chemotherapy?) Some of the differences may seem remarkable without this context, for example, it appears that ~50% of relapse samples in this category have amplification of the AKT1 gene vs. 0% of the diagnosis samples. Is this just a fluke due to low numbers?

Response: We have improved figure presentation of S15 (now S13) to exclude non-informative comparisons (prior lines > 3), included colour codes in legends, and added sample counts to plots throughout the manuscript.

-
- Figure 4B. A p.value should be added to the boxplots.

Response: We have added p-values for the comparisons in Figure 4B as requested

-
- Figure S23. This figure is complex and without a statistical analysis highlighting the results worth noticing, it is very hard to interpret.

Response: We acknowledge the criticism of this figure. Given that we do not directly reference any subplots in this figure we have now removed it.

-
- Figure 5A. Please add 'NS' to indicate that the comparisons are not significant, if appropriate.

Response: We have added p values to Figure 5A, which may be more informative than "NS" labels

-
- Figure 5C. Without a p.value label, it is hard to recognize the shift in s5 as the significant feature in the figure (i.e., the shift in s1 seems just as pronounced). Also, since the authors eventually refute the biological nature of this shift, Figure 5C should be relegated to the supp data.

Response: We have improved the caption and added a p value to the radar plot (Wald test) showing significant differential abundance of signatures between diagnosis and recurrent samples with and without signature 5, though we still support the inclusion of 5C within figure 5.

-
- Figure S24. Please add the significant p.values to the corresponding graphs.

Response: Brackets highlighting the significant comparisons have been added to Figure S24 (now figure S18) as well as an overhaul of the presentation to improve consistency. Additionally, outstanding clinical data queries have been addressed, which has allowed us to increase the number of samples included in the tissue-specific analysis, with improved statistical power. However, this has not resulted in any significant changes to the results.

• Figure 7. The term ‘non-primary resistant’ in the legend is confusing (same as in Fig. S31). Are these primary (i.e., at diagnosis) tumors that are not resistant, resistant tumors that are not primary, or any tumors that do not show primary resistance? I believe the first definition is correct, but it warrants clarification. Maybe better terms would be ‘tumor with/without primary resistance’. Also, since primary resistance is defined as relapse <6 months after completion of first-line treatment, is it appropriate to classify relapse tumors as primary resistant, or should the distinction be limited to diagnosis tumors?

Response: We thank the reviewer for this comment. In these analyses, we compared diagnosis samples from patients who subsequently relapsed within six months of completion of first-line chemotherapy with diagnosis samples from all other patients. To improve clarity, we have changed the names of these two groups to ‘primary platinum resistant’ and ‘others’ respectively.

• Figure 7C, D and E. These graphs need some more detailed explanations. The text simply reports that ‘patients with primary platinum resistance also had significantly different CN signature exposures at diagnosis with significantly lower exposure to s3 and higher s6 exposure than samples from all other patients’. While Fig. 7C does show these significant differences, they are not obvious in Fig. 7E (where s1 seems to show a larger delta) and it is unclear what Fig. 7D represents. Also increased s1 in patients with primary resistance is reported in the discussion but not in this corresponding result section, which adds to the confusion.

Response: We thank the reviewer for this comment. We have corrected a misreporting of s6 as being of note in this results section instead of s1, which was intended. In order to improve the clarity of presentation, we have reduced figure 7 to only three panels (A-C) (now main figure 6) and added additional points of clarification to the figure caption.

• Figure S31B. Please double check that the violin plots match the number of samples reported in the legend.

Response: Thank you. We have updated the figures and captions to show correct sample counts for figure S31B (now S26B).

Reviewer #2: ovarian cancer treatment resistance and clinical expertise

This report presents an extremely valuable real-world dataset of genomic analyses of initial tumor samples and biopsies at relapse from women with ovarian cancer. Surprisingly, the investigators identify limited differences in genomic parameters at the time of relapse. Overall this work is hypothesis generating, and it builds on prior analyses of ovarian cancer cohorts as described in the discussion. In particular, the identification of markers associated with primary platinum resistance present an opportunity to stratify patients for frontline treatment if validated clinically. Additional strengths of the study include the

relatively tight timeframe during which patients were recruited and secondary samples obtained, the heterogeneity of treatment regimens that reflect standard clinical practice, and the paired analyses enabling an assessment of correlations between immune parameters and genomic alterations as well as differences in tumour profiles for individual patients. The inclusion of substantial supplementary data - in particular individual patient vignettes - creates a resource that will continue to support research efforts in the field.

Response: We thank the reviewer for these positive comments.

Additional information that would be helpful to guide interpretation of the data includes:

- Clarifying the specific make-up of subgroups analyzed for different parameters. Although 276 patients were enrolled, the analyses included much smaller cohorts ranging from 21 to 134 subjects. A table is provided in the supplemental data that tracks the number of patients included in each analysis but it's not clear why samples from only 47 patients were available for paired CN analysis or whether these subgroups differed significantly from the entire study cohort with respect to clinical characteristics.

Response: Although we recruited 276 patients, it was not possible to perform all analyses on all samples owing to variations in DNA quantity and quality. We have clarified the number of cases included in all analyses in the figure captions and included a new table in the Supplementary Methods to clarify how many samples were included in each major analysis.

-It would be helpful to have a separate statistical methods section included.

Response: All statistical methods are presented in the Supplementary Methods section as some analyses are complex and use hypothesis-specific methods, which cannot be easily summarised in a short methods section.

- Survival curves are included but this is not an interventional trial. With substantial differences (>10 yrs) in the time since diagnosis as illustrated in Fig 1B, a comparison of outcomes is difficult to interpret.

Response: Time since diagnosis is not typically a consideration when presenting outcomes as Kaplan Meier plots in studies in relapsed HGSC. The purpose of the Kaplan Meier plots in Figure 1B is simply to demonstrate the outcomes of the patients in BriTROc-1. The graphs suggest that the outcomes are broadly in line with previously-published cohorts of patients with platinum-sensitive and platinum-resistant disease, indicating that our patients are likely to be representative of real world recurrent populations in standard oncology practice.

- The authors acknowledge the potential for bias in selecting patients healthy enough for biopsy/secondary debulking. It would be helpful to additionally note whether subjects were already scheduled for biopsy or secondary debulking by their treating oncologist, or whether this was done specifically for the purposes of this study.

Response: The only study-mandated intervention in BriTROc-1 was a biopsy, and patients received standard of care therapy determined by their oncology team. Patients were eligible for BriTROc-1 regardless of whether secondary debulking surgery was planned (the study recruited before the results of

the DESKTOP-3 study were available). We include the method for obtaining tissue in the manuscript: of the 259 patients who underwent a study-entry biopsy, 142 had either CT or ultrasound guided biopsies, whilst 111 had biopsies taken during some form of interventional procedure.

- While differences in fixation methods are discussed as potential confounding factors for analyses, the age of the samples is not considered. Were older samples more likely to fail QC? Did this skew the cohorts analyzed?

Response: Thank you for raising this important question. We have analysed both cohorts (diagnosis and relapse) in both sWGS and SNV analyses and found minimal differences in sample age relating to failed QC (see rebuttal figures below). In the sWGS samples, relapse samples that failed QC were significantly *younger* than those that passed but we do not believe this would have added any significant bias to the study.

For the TAm-Seq libraries derived from DNA samples, we also observed that samples which failed QC were significantly *younger*, though this time for the diagnosis samples (see figure below). Again, we do not believe this would have added any significant bias to the study.

- The title of the paper specifies HG serous cancers, but pts with non-serous cancers were also enrolled. Fig S1 suggests these cases were specifically excluded from some analyses. A justification for these decisions would be helpful to include in the methods.

Response: We thank the reviewer for raising this point. Patients with carcinosarcoma and grade 3 endometrioid carcinoma were eligible for recruitment in BriTROc-1, but only seven such patients were enrolled (Table 1) and they were not excluded from any analysis. However, we acknowledge that the wording on Figure S1 was confusing - the samples that were removed were those with low cellularity. We have amended Figure S1 accordingly.

- The individual vignettes of patients is a particularly rich source of information but a legend would be helpful - to explain the dashed red lines on the figures, to specify which biopsy was used for this study, etc. In addition, some cases appear to be missing data. For example pt 147 is classified as platinum resistant but she was treated with platinum-based chemotherapy after biopsy. Patient 132 had two debulking surgeries and a biopsy but it's not clear which sample(s) were included as the relapse sample. Finally, several patients (157, 232, 187) had recurrence diagnosed but no treatment started for several months. This is surprising and may indicate there is missing treatment information.

Response: Thank you for highlighting this. The individual vignettes have been updated to include an improved legend and sample labelling to clarify which sample for a given patient is plotted. We have the standard clinical definitions of platinum-sensitive and platinum-resistant, but all treatment of patients in BriTROc-1 was at the discretion of treating oncologists. Patients who have relapsed but who are asymptomatic can undergo periods of surveillance before commencing chemotherapy.

Minor:

- Fig 1A has an arrow directed from the relapse figure to the figure at diagnosis? Fig 1

Response: The arrow directed reflected the fact that BriTROc-1 recruited patients at relapse and utilised archival material from diagnosis. However, we have now corrected the arrow direction to match disease progression rather than sample recruitment.

Reviewer #3 (Remarks to the Author): expertise in statistical analysis of clinical trial data

The genomic landscape of recurrent ovarian high grade serous carcinoma: the BriTROC-1 study”

The manuscript reported the results from the BriTROC-1 study which aims to investigate genes that are associated with recurrence and resistance in ovarian high grade serous carcinoma. The study is well designed, with its sequencing data gone through appropriate bioinformatic pipelines and statistical analysis. To seek genomic signal associated with innate and acquired resistance, the study investigated genome-wide mutation, copy number, selected genes and gene signatures on population level as well as on individual patient level (paired samples). A throughout set of analysis have been carried out, including comparisons guided by various prognostic clinical features, pathway-level enrichment analysis, and clustering to explore hidden patient sub-cohorts. I am convinced by the conclusion that HGSC has a stable genome, and it is unlikely the cause of acquired platinum resistance.

Comments:

1. The abstract needs more key numbers. I understand that the study did not find much genomic changes between diagnosis and relapse samples but the phrase such as “very strong concordance” is too vague for the readers to capture the key messages of the study.

Response: Unfortunately, the abstract is limited to 150 words, and we have included as much detail as possible.

2. The study included various data types and analysis, each comprised different numbers of patient samples. While the authors mentioned that the limited sample sizes are weak points of some analysis, I found it hard to keep track on sample sizes during reading. Please make sure that sample size information is properly included for each analysis in the main text, as well as corresponding figures, supplementary figures, and tables. One example would be on line 324, where the $p=0.02$ was achieved on a comparison between 7 patients and 19. Unfortunately these small sample sizes were not mentioned in the main text.

Response: Thank you for raising this important comment. We updated our figure captions to state the numbers of samples analysed in each figure and included a table in the Supplementary Methods detailing the sample numbers in all comparative analyses.

3. In line 207, I can not align the mutation events reported in Fig 2 with those in Fig S4, S5. Please double check.

Response: We thank the reviewer for this comment. We have removed original Figures S4 and S5 and give a fuller explanation of somatic variant calling in the Supplementary Methods, which explains minor differences in some analyses depending on the availability of matched germline DNA.

4. Line 327, please describe clearly which site has significantly higher S1, as it is not obvious from visual in the Figure S24D.

Response: Brackets highlighting statistically significant comparisons have been added to Figure S24 (now S18). We have also made other improvements to the figure presentation to improve consistency. Additionally, outstanding clinical data queries have been addressed, which has allowed us to increase the

number of samples included in the tissue-specific analysis, with improved statistical power. However, this has not resulted in significant changes to the results.

5. Line 375, the association between *CCNE1*/*KRAS* amplification and the binary status of platinum sensitivity was assessed. Can you also show the association between the genes and time to relapse to enable better understanding of the relationship between the genes and resistance?

Response: Thank you. The analysis that the reviewer mentions relates to diagnosis samples from patients who subsequently relapsed within six months of end of first line platinum chemotherapy (now consistently termed 'primary platinum resistant' throughout the MS) and diagnosis samples from all other patients (now called 'other'). Thus, the presence of *KRAS* and *CCNE1* amplification at the time of diagnosis is significantly associated with primary platinum resistant relapse.

6. Line 378, please double check the colour code of Figure S31-B, as the results reported in the main text can not be aligned with the figure.

Response: Thank you. The colour coding for the 'primary platinum resistant' vs 'others' comparison is now consistent between Figures 7 (now figure 6) and S31 (now figure S26).

REVIEWERS' COMMENTS

Reviewer #1 (Remarks to the Author):

The authors addressed all of my comments satisfactorily. I have no further comments.

Reviewer #2 (Remarks to the Author):

The revisions made by the authors have addressed prior concerns and significantly clarified the report.

Reviewer #3 (Remarks to the Author):

The authors have addressed all my comments to a satisfactory level. Overall the manuscript reads better and is ready to be published.